# Improving Multimodal Protein Function Prediction Using Bidirectional Interaction and Dynamic Selection Mechanisms

## Abstract

Protein function prediction is pivotal for uncovering the mechanisms of life processes. Protein function prediction is a multi-label classification task with numerous functional labels that exhibit hierarchical relationships. Relying solely on unimodal protein features is insufficient for computational models to capture complex protein functions adequately. Recently, several methods for protein function prediction have enhanced the performance by integrating multimodal protein features. However, since multimodal protein features describe protein functions from different perspectives, it is challenging to capture the intricate relationships among these multimodal features with different meanings and heterogeneity. Therefore, we propose a multimodal method for protein function prediction that can effectively utilize the intricate internal relationships between spatial structure features (i.e., protein-protein interaction network, subcellular location, and protein domains) and sequence features (i.e., amino acid sequence). In this work, we introduce the Bidirectional Interaction Module (BInM) to facilitate interactive learning between multimodal features by mapping spatial structure and sequence features of proteins to each other. Moreover, to deal with the difficulty of hierarchical multi-label classification in this task, a multi-branch Dynamic Selection Module (DSM) is designed to select the feature representation that is most favorable for current protein function prediction. Comprehensive experiments on human datasets demonstrate that our model outperforms state-of-the-art multimodal-based methods such as Graph2GO, DeepGraphGO, and CFAGO. Furthermore, we assess the efficacy of the features through Davies-Bouldin scores and t-SNE visualization experiments. The experimental results show that our method constructs more useful protein representations through bidirectional interaction and dynamic selection mechanisms, leading to improved accuracy in protein function prediction. The code in this work will be made public after its acceptance.

## 1 Introduction

Proteins, as essential components of life, play a crucial role in biological research. With the rapid development of bioinformatics (Giamarellos-Bourboulis et al., 2024; Hasselgren & Oprea, 2024), protein function prediction has emerged as a key challenge in the field of biology. Protein functions are standardized through the Gene Ontology (GO) framework. This framework classifies protein functions into three categories: biological process ontology (BPO), molecular function ontology (MFO), and cellular component ontology (CCO)(Aleksander et al., 2023). In recent decades, numerous deep learning-based computational methods (You et al., 2021; Zhang et al., 2023) have been developed to predict protein functions. Most of the previous methods (Kulmanov & Hoehndorf, 2020) utilize one of the following types of information: sequence information, structure information, and protein-protein interaction (PPI) network. In the process of analyzing each type of protein information (Kulmanov & Hoehndorf, 2020), we found that relying on a single-modal feature to predict protein function is often constrained by the conditions of the data itself. For instance, many studies (Fan et al., 2020) have shown that using protein sequence information significantly improves the accuracy of molecular function predictions. However, there are many proteins that share functional similarities but have dissimilar sequences (Lin et al., 2024). As a result, for proteins with

low sequence similarity, the accuracy of predictions may be compromised. Moreover, structure-based methods (Jiao et al., 2023; Gligorijević et al., 2021) leverage the rich structure information provided by resources to improve protein function prediction accuracy. But the high complexity of protein structures and the cost of data acquisition limit the application of structure-based methods (Paysan-Lafosse et al., 2023). Furthermore, the noise introduced during the generation of PPI networks through high-throughput techniques poses risks to the accuracy of predictions (Chen & Luo, 2024). Therefore, integrating these different types of protein data based on multimodal methods and taking advantage of their complementary advantages in functional prediction is an important way to improve the performance of protein function prediction.

Recognizing that unimodal representations are insufficient to encapsulate the information contained within proteins, multimodal-based methods have emerged. DeepFRI (Gligorijević et al., 2021) leverages graph convolutional networks to learn features from both protein sequences and structural properties. Graph2GO (Fan et al., 2020)utilized graph networks to consolidate sequence similarity networks and PPI networks, incorporating protein sequence and structural information as node features for function prediction. However, those using GNNs may amplify noise and face issues with over-smoothing. To address these limitations, CFAGO(Wu et al., 2023) proposed the incorporation of Transformer mechanisms within autoencoders to fuse multimodal protein features.

However, current multimodal approaches primarily rely on information fusion mechanisms without considering the potential complementarity between different modalities. To address this issue, we propose a bidirectional-interaction and dynamic-selection-driven method (BDGO) that integrates spatial structure information (i.e., PPI network, subcellular location, and protein domains) and sequence information (i.e., amino acid sequence) from proteins. In addition, large language models play an important role in improving protein function prediction. SaProt(Su et al., 2023), as a large-scale general-purpose protein language model (PLM) trained on 40 million protein sequence and structure data, achieved good results in protein function prediction tasks. Inspired by large language models, the protein sequence information in our method is extracted using the pre-trained ProtT5 (Elnaggar et al., 2021). In this work, to better learn multimodal information, our proposed BDGO model includes a shared learning branch and an interactive learning branch. In the shared learning branch, we concatenate features from different modalities and perform joint analysis in a unified representation space. Moreover, we introduce the Bidirectional Interaction Module (BInM), which means that each modality not only influences the processing of other modalities but also obtains information from them, thereby enhancing the overall understanding capability.

Further, faced with thousands of protein functions, accurately predicting the protein function of a sample remains a challenging issue. Protein function prediction is essentially a complex hierarchical multi-label classification problem. In this situation, we propose the Dynamic Selection Module (DSM) to dynamically select the optimal feature combination for fitting more diverse protein functions. Our main contributions can be summarized as follows:

- We propose a multimodal feature-based approach for protein function prediction that overcomes the limitations of single-modality methods, effectively representing protein functional characteristics to assist the model in understanding protein function.

- Our proposed BInM incorporates a bidirectional interaction mechanism to promote efficient fusion and information exchange between sequence features and spatial features, enhancing the model's ability to capture strong protein information between different modes.

- We design the DSM that enables the model to adaptively select channel features most relevant to specific functional labels, resulting in enhanced classification performance.

## 2 METHODOLOGY

Our proposed method efficiently captures multimodal information of proteins through a strategy for two-step training. In the pre-training stage, we use the encoder-decoder model to learn and inject multimodal knowledge. For spatial features including PPI, subcellular location, and protein domains, a Protein Spatial Structured Information (PSSI) encoder-decoder model using the BiMamba blocks is introduced in this stage. To mine sequence features including protein sequences, we design a Protein Sequence Information (PSI) encoder-decoder model based on the Transformer blocks for pre-training. Then, during our BDGO model training phase, we integrate and learn features from

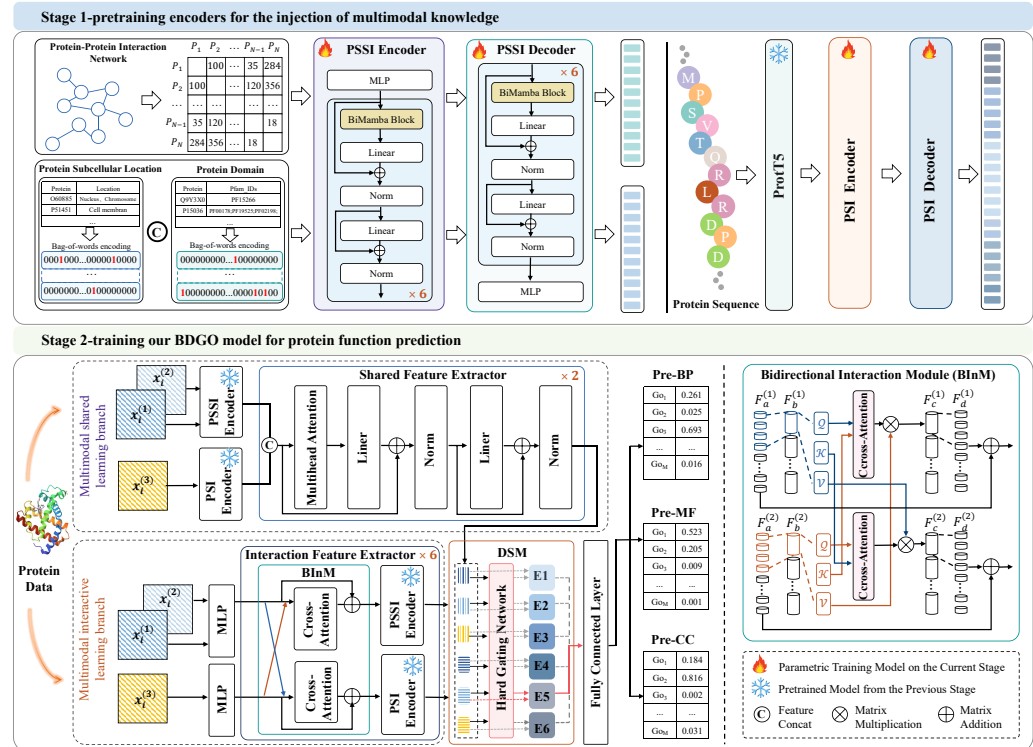

Figure 1: An illustration of our proposed method. This method is mainly divided into two stages. The first stage is to pretrain the Protein Spatial Structure Information (PSSI) encoder and Protein Sequence Information (PSI) encoder for the injection of multimodal knowledge . The second stage is training our proposed BDGO model, which consists of an MSL-Branch, a MIL-Branch with the Bidirectional Interaction Module (BInM), and the Dynamic Selection Module (DSM).

multimodal information. The proposed model is primarily divided into two major branches: one is the multimodal shared learning branch (MSL-Branch), and the other is the multimodal interactive learning branch (MIL-Branch). Protein data are processed through these multiple branches to generate several sets of features, which serve as inputs for our well-designed hard gating network. Finally, the model dynamic selects the optimal features for the current protein, to enhance performance in protein function prediction. An illustration of our proposed method can be seen in Figure 1.

## 2.1 ENCODER-DECODER PRETRAINING

### 2.1.1 PROTEIN SPATIAL STRUCTURE INFORMATION (PSSI) ENCODER-DECODER

The PPI network gets an $N \times N$ adjacency matrix by matrix conversion as input to the encoder. Moreover, another input to the encoder is obtained by concatenating the bag-of-words encodings of subcellular location and Protein Domain.

**Mamba Preliminaries.** Mamba (Gu & Dao, 2023) extends the capabilities of the State-Space Models (SSMs) (Gu et al., 2023) by enabling the transformation of a continuous 1D input $x_t \in \mathbb{R}$ to $y_t \in \mathbb{R}$ via a learnable hidden state $h_t \in \mathbb{R}^{\hat{N}}$ with discrete parameters $\bar{A} \in \mathbb{R}^{\hat{N} \times \hat{N}}$, $\bar{B} \in \mathbb{R}^{1 \times \hat{N}}$, and $\bar{C} \in \mathbb{R}^{1 \times \hat{N}}$ as follows:

$$h_t = \bar{A}h_{t-1} + \bar{B}x_t, \ y_t = Ch_t + Dh_t, \ \bar{A} = e^{\Delta A}, \ \bar{B} = (\Delta A)^{-1}(e^{\Delta A} - I) \cdot \Delta B, \ \bar{C} = C. \tag{1}$$

$\bar{A}$ and $\bar{B}$ are continuous $A$ and $B$ converted to discrete evolution parameters using a timescale parameter $\Delta$. To process discrete-time sequences that are sampled at intervals of $\Delta$, SSMs can be calculated using the recurrence formula. $\bar{C}$ represents the projection parameters. In addition, the

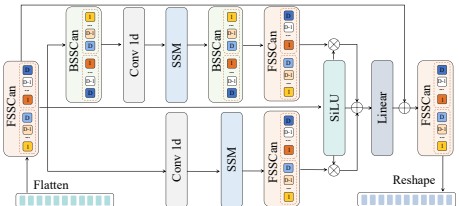

Figure 2: Structure of the BiMamba block.

models compute output through a global convolution as in the following:

$$\bar{K} = (\bar{C}\bar{B}, \bar{C}\bar{A}\bar{B}, \ldots, \bar{C}\bar{A}^{\hat{N}-1}\bar{B}), \ y = x * \bar{K}, \tag{2}$$

where $\hat{N}$ is the length of the 1D input $x$, and $\bar{K}$ is a structured convolutional kernel.

**BiMamba Block.** Inspired by the selective scan mechanism in Vision Mamba (Zhu et al., 2024), BiMamba Block introduces a novel bidirectional selective scanning mechanism designed for protein data, capturing both the start and end of spatial structure features for enhanced detail and context. Multi-dimensional features are first converted into one-dimensional vectors. Features $x_{sp}$ from PPI, subcellular location, and protein domains are then passed through BiMamba blocks, interleaved with linear layers and residual operations. As shown in Fig. 2, forward (FSScan) and backward selective scans (BSScan) extract bidirectional matrix features via positional transformations and reconstructions. Transformed tokens are scanned using Equation 1 to produce new features, with BiMamba's output $\tilde{x}_{sp}$ expressed as:

$$\tilde{x}_{sp} = FSSCan(x_{sp}) + FSSCan(Linear(F_\alpha \odot F_\sigma + F_\beta \odot F_\sigma + F_\sigma)), \tag{3}$$

$$F_\alpha = FSSCan(BSSCan(SSM(Conv\,1d(BSSCan(FSSCan(x_{sp})))))), \tag{4}$$

$$F_\beta = FSSCan(SSM(Conv\,1d(FSSCan(x_{sp})))), \tag{5}$$

$$F_\sigma = SiLU(FSSCan(x_{sp})), \tag{6}$$

where the operation $\odot$ denotes the Hadamard product.

**PSSI Encoder.** In this section, we propose a PSSI encoder architecture designed to effectively map high-dimensional input data into a low-dimensional latent space. The PSSI encoder is composed of multiple neural network layers, including multilayer perceptrons (MLPs), BiMamba block, Linear and Norm layers, which work in concert to extract features from the input data and generate a compact latent representation. Assume that the input feature $x_i^{h(k)} \in \mathrm{R}^{H_i^k}$ is a high-dimensional vector of the $i$-th protein, and it is reconstructed utilizing the MLP layer. Then the reconstructed features are processed by the PSSI encoder to output a low-dimensional representation $x_i^d(k) \in \mathrm{R}^{D_i^k}$.

**PSSI Decoder.** The architecture of the PSSI decoder is a counterpart to that of the encoder. The PSSI decoder rebuilds the given protein spatial structure information based on the hidden representations output by the encoder. This process involves BiMamba computation and residual operations, optimizing the cross-entropy loss function to enhance the performance. After taking the output $x_i^d(k)$ of the PSSI encoder and passing through the BiMamba block, alternating Linear and Norm layers, we obtain the recovered high-dimensional features $\bar{x}_i^h(k) \in \mathrm{R}^{H_i^k}$.

The overarching objective of the encoder-decoder architecture is to minimize the sample wise binary cross-entropy loss between the original and reconstructed source features, thereby enhancing the model's predictive accuracy and fidelity in representing complex protein data. The loss function of PSSI encoder-decoder is:

$$\mathcal{L}_{sp} = \frac{1}{N} \sum_{i=1}^{N} \sum_{k=1}^{K} \sum_{j=1}^{H_i^k} -\left[ x_{ij}^{h(k)} \log \bar{x}_{ij}^{h(k)} + \left(1 - x_{ij}^{h(k)}\right) \log \left(1 - \bar{x}_{ij}^{h(k)}\right) \right], \tag{7}$$

where $N$ is the number of total proteins, $K$ is the number of input sources, $H_i^m$ is the feature dimension of the $k$-th source, $x_{ij}^{h(k)}$ denotes the $j$-th dimension vector of the input feature $x_i^{h(k)}$, and $x_i^{h(k)}$ represents the $j$-th dimension vector in generated feature $\bar{x}_{ij}^{h(k)}$.

### 2.1.2 PROTEIN SEQUENCE INFORMATION (PSI) ENCODER-DECODER

In PSI encoder-decoder, transformer block with multi-head self-attention mechanism (Dosovitskiy et al., 2021) is used to extract the long-distance features of the protein sequences. Particularly, to fully exploit the protein sequence features, we use pre-trained ProtT5 (Elnaggar et al., 2021) model to parse the protein sequences. To achieve this, we froze the parameters of ProtT5 and connected it to our PSI encoder for further pretaining.

**PSI Encoder.** The PSI encoder consists of an MLP block and 6 self-attention blocks. The self-attention block includes a multi-head self-attention (MSA) computation layer, as well as alternating linear and norm layers, connected through a residual structure. Assuming the input feature to the self-attention block is $\tilde{s}_i^d = MLP(s_i^h)$, the output feature is $\hat{s}_i^d \in \mathrm{R}^{D_i}$:

$$\hat{s}_i^d = N(N(\tilde{s}_i^d + L(MSA(\tilde{s}_i^d))) + L(N(\tilde{s}_i^d + L(MSA(\tilde{s}_i^d))))), \tag{8}$$

where $s_i^h \in \mathrm{R}^{H_i}$ is the $i$-th input sequence feature of encoder, $L(x)$ denotes the fuction of Linear layer, and $N(x)$ denotes the Norm layer.

**PSI Decoder.** The PSI decoder takes the hidden states from the encoder as input, which contains compressed information about the input sequence. To obtain the final protein sequence encoding, we designed the PSI decoder using a combination of 6 self-attention blocks and one MLP block. Then, the output feature of the PSI decoder is $\hat{s}_i^h \in \mathbb{R}^{H_i}$. Like the PSSI encoder-decoder, the loss function $\mathcal{L}_{se}$ for the PSI encoder-decoder also adopts the form of cross-entropy:

$$\mathcal{L}_{se} = \frac{1}{N} \sum_{i=1}^{N} \sum_{j=1}^{H_i} - \left[ s_{ij}^h \log \bar{s}_{ij}^h + \left(1 - s_{ij}^h\right) \log \left(1 - \bar{s}_{ij}^h\right) \right], \tag{9}$$

where $i$ denotes the sequence input of the $i$-th protein, $j$ is the $j$-th dimension vector of the feature map, and $H_i$ is the dimension of input feature.

## 2.2 BDGO MODEL

### 2.2.1 BIDIRECTIONAL INTERACTION MODULE (BInM)

The proposed BInM enhances the model's ability to learn complex patterns by integrating information across modalities. Using dual-branch cross-attention, it compares query (Q) vectors with key (K) vectors from the opposite branch, enabling bidirectional interaction. This approach captures interdependencies between branches more effectively, similar to multi-head self-attention but focused on cross-branch connections.

Therefore, we assume that the features transformed by PPI are represented as $x_i^{(1)}$, and the features obtained from the encoding of subcellular location and protein domains are concatenated to form $x_i^{(2)}$, while the features extracted through the ProtT foundation model for protein sequences are denoted as $x_i^{(3)}$. Subsequently, $x_i^{(1)}$ and $x_i^{(2)}$ get features with the same dimension after the MLP block reconstruction features, and their concatenated feature map $\widetilde{x}_i^B$ is used as the input of the first branch of BInM. Similarly, the input $\overline{x}_i^B$ to the second branch of BInM is obtained through the MLP block. In BInM, the input embedded patches $F_a^{(1)} \in \mathbb{R}^{L_a \times D_a}$ and $F_a^{(2)} \in \mathbb{R}^{L_a \times D_a}$ are initially and randomly divided into multiple heads vectors $F_b^{(1)} \in \mathbb{R}^{L_a \times D_b \times H_b}$ and $F_b^{(2)} \in \mathbb{R}^{L_a \times D_b \times H_b}$, where $H_b$ is the number of multiple heads.

As shown in Figure 1, $F_b^{(1)}$ and $F_b^{(2)}$ are converted into queries $\mathcal{Q}^{(1)}(F_b^{(1)})$ and $\mathcal{Q}^{(2)}(F_b^{(2)})$. The key $\mathcal{K}^{(1)}$ and value $\mathcal{V}^{(1)}$ of $F_b^{(1)}$, and the key $\mathcal{K}^{(2)}$ and value $\mathcal{V}^{(2)}$ of $F_b^{(2)}$ are obtained using three generators $\mathcal{Q}$, $\mathcal{K}$, and $\mathcal{V}$. Then, $F_c^{(1)} \in \mathbb{R}^{L_a \times D_b \times H_b}$ obtained by cross-attention is defined as:

$$F_c^{(1)} = softmax(\mathcal{Q}^{(1)}(F_b^{(1)}) \otimes \mathcal{K}^{(2)}(F_b^{(2)})^T) \otimes \mathcal{V}^{(2)}(F_b^{(2)}), \tag{10}$$

where the operation $T$ means matrix transpose, the operation $\otimes$ represents matrix multiplication, and the goal of $softmax$ function is to normalize the $F_c^{(1)}$. Finally, the cross-attention output feature $F_d^{(1)} \in \mathbb{R}^{L_a \times D_a}$ of the first branch is obtained by feature mapping. Similarly, we can get the cross-attention output $F_d^{(2)} \in \mathbb{R}^{L_a \times D_a}$ of the second branch. In this way, the model takes into account not only the meaning of each branch itself, but also the relationships with other branch features, resulting in a richer and more accurate representation on multimodal data.

### 2.2.2 DYNAMIC SELECTION MODULE (DSM)

In the final feature selection stage, we introduce DSM to enhance key features and mitigate the impact of conflicting ones. As illustrated in Figure 1, this module employs a Mixture-of-Experts (MoE) (Masoudnia & Ebrahimpour, 2014) strategy for dynamical feature selection. The features extracted by the MSL and MIL branches are combined into the input of DSM, denoted as $x_{dsm} = (x_{dsm}^1, x_{dsm}^2, \cdots, x_{dsm}^V)$, where $V$ is the number of expert networks $E(x)$, each responsible for processing one group of features. We set up a hard gating network $G(x)$ to decide which expert should be activated. Unlike traditional MoE systems, which combine outputs from all experts through weighted averaging, our hard gating network selects a single expert for computation. This approach allows the model to better adapt to the complex and large-scale protein function prediction tasks. The hard gating network is composed of two Linear layers. Inputting $x_{dsm}$ into $G(x)$ for computation yields a $V$-dimensional one-hot decision vector $g = one - hot(\arg\max_v G(x)_v)$. Finally, the output of DSM is $x_\epsilon = \sum_{v=1}^V g_v E_v(x_v)$, where $x_v$ represents the $v$-th groups of the inputted feature of DSM.

### 2.2.3 PROTEIN PREDICTION

In this work, protein function prediction is modeled as the multi-label classification task. The output feature $x_\epsilon$ of the DSM is used as input to the predictor, which is constructed from fully connected layers. The predictor outputs a score vector of $M$-dimension GO terms $P_i = (p_i^1, p_i^2, \cdots, p_i^M))$.

**Loss Functions.** In the context of GO terms, there are significantly more negative proteins than positive ones in the training set. Consequently, we employ an asymmetric loss (Wu et al., 2023) as the prediction loss $\mathcal{L}_{pre}$. The loss function $\mathcal{L} = \mathcal{L}_{pre} + \mathcal{L}_{gate}$ of the final model consists of the loss of the prediction and the loss of the gating network.

$$\mathcal{L} = \frac{1}{NM} \sum_{i=1}^N \sum_{m=1}^M -y_i^m \left(1 - p_i^m\right)^{y+} \log\left(p_i^m\right) - \left(1 - y_i^m\right) \left(p_i^m\right)^y - \log\left(1 - p_i^m\right) + \lambda \sum_{v=1}^V g_v C\left(E_v\right),$$

(11)

where $y_i^m$ represents the ground truth label for the $i$-th protein, while $p_i^m$ denotes the predicted score. The symbols $\{y+\}$ and $\{y-\}$ refer to the positive and negative focusing parameters respectively. $C(E_v)$ denotes the running cost of the $v$-th expert in DSM.

## 3 EXPERIMENTS

In this section, we present the experimental setup, including the datasets, baseline models, training details, and evaluation metrics. Then we provide an analysis of the experimental results, supported by ablation studies and Davies-Bouldin scores to validate the effectiveness of the model.

### 3.1 EXPERIMENTAL SETUP

**Dataset.** We construct our dataset with reference to CFAGO(Wu et al., 2023). The PPI data is obtained from the STRING (Szklarczyk et al., 2023) database (version 11.5). Protein sequences, subcellular localization, and domain data are collected from the UniProt (Consortium, 2022) database (version 3.5.175). A total of 19,385 proteins are used for pretraining. For the fine-tuning dataset, we first collected protein function annotation data from the Gene Ontology (Aleksander et al., 2023) Resource database (version 2022-01-13). Following the standards of the CAFA (Radivojac et al., 2013)challenge, we extracted GO terms with evidence codes (EXP, IDA, IPI, IMP, IGI, IEP, TAS, and IC) as labels. Proteins annotated with these GO terms in the pretraining dataset were selected as the fine-tuning dataset. The dataset was then split based on two time points. The finetuning dataset for each GO branch is organized as follows: BPO includes 3,197 training proteins, 304 validation proteins, and 182 testing proteins. MFO includes 2,747 training proteins, 503 validation proteins, and 719 testing proteins. CCO includes 5,263 training proteins, 577 validation proteins, and 119 testing proteins. Additionally, the number of GO terms is 45 for BPO, 38 for MFO, and 35 for CCO. We further provide the similarity distributions of the test sets and the corresponding model performance in Appendix Section 6.3.

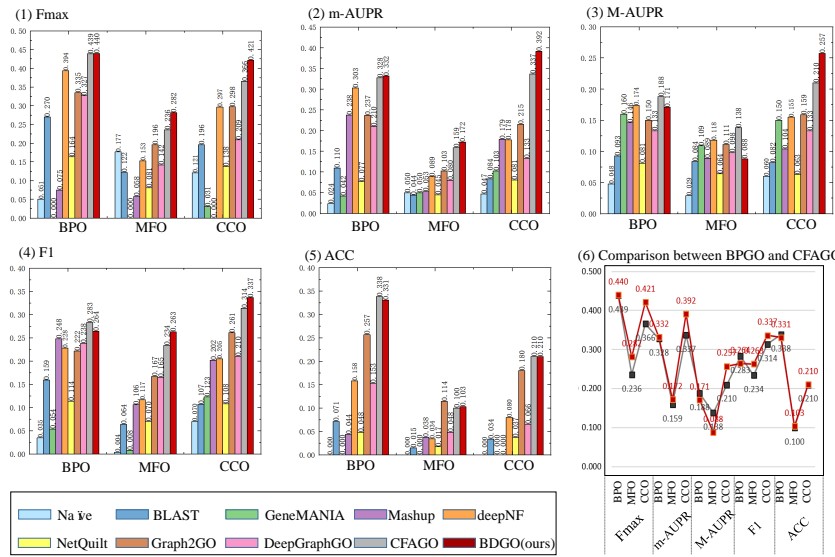

Figure 3: Performance comparison of different computational methods.

**Implementation Details.** All methods are implemented by PyTorch, and we conduct all experiments on a single NVIDIA GTX 4090 GPU with 24GB of memory. The batch size is set to 32. Additionally, we combine the training and validation sets to train our model. We set the dropout rate to 0.1 during pre-training, and the model trains for 5000 epochs, with a learning rate of $1 \times 10^{-5}$ for the first 2500 epochs and $1 \times 10^{-6}$ for the remaining 2500 epochs. During fine-tuning, different hyperparameters are used for each aspect, with learning rates set to 3.6e-4, 8.6e-2, and 1.4e-4 for BPO, MFO, and CCO, respectively. The same pre-trained model serves as the feature extractor, and AdamW is used as the optimizer across all aspects. For BInM module, the cross-attention mechanism is configured with 8 heads, while other settings follow the default parameters in torch.nn. In DSM module, the temperature parameter $\tau$ is set to 1.

**Compared Methods.** We compare BDGO with nine methods. Based on the data types used by each method, we roughly divide the nine baseline methods into three types: sequence-based methods (Naive (Radivojac et al., 2013), BLAST(Altschul et al., 1990)), PPI network-based methods (GeneMANIA(Mostafavi et al., 2008), deepNF(Gligorijević et al., 2018), Mashup(Cho et al., 2016), NetQuilt(Barot et al., 2021)), and multimodal methods (Graph2GO(Fan et al., 2020), Deep-GraphGO(You et al., 2021), CFAGO(Wu et al., 2023)). All methods are trained on single-species datasets using the hyperparameters and network architectures reported in the corresponding papers, and all results undergo five random repetitions for validation.

**Evaluation Metrics.** In this study, we evaluate the predictive performance of various methods using five metrics, offering different perspectives on model accuracy and effectiveness. These include two types of area under the precision-recall curve (AUPR)(Davis & Goadrich, 2006): micro-averaged AUPR (m-AUPR) and macro-averaged AUPR (M-AUPR) (Peng et al., 2021), as well as the F1-score (F1)(Wu et al., 2023), accuracy (ACC), and F-max score ($F_{max}$)(Lin et al., 2024).

### 3.2 COMPARISON WITH UNIMODAL-BASED AND MULTIMODAL-BASED METHODS

As shown in Figure 3 and Table 1, BDGO outperforms other methods across multiple metrics in all three domains. It achieves the best performance in two key metrics: Fmax and m-AUPR, particularly in MFO and CCO. Specifically, BDGO reaches the highest Fmax values of 0.282 in MFO and 0.421 in CCO, representing improvements of 19.5% and 15.0% over the current state-of-the-art, CFAGO (0.236 and 0.366). Additionally, BDGO achieves m-AUPR values of 0.172 in MFO and 0.392 in CCO, which are 8.2% and 16.3% higher than CFAGO (0.159 and 0.337). These results demonstrate the significant advantage of BDGO in single-species protein function prediction. In addition, we discuss the Structure-based and PLM-based comparison methods, as detailed in Appendix 6.6.

Table 1: Comparison results of different methods. The best results are highlighted in bold, and the sub-optimal results are underlined. After the ± is the standard deviation of the experimental results.

| Method | | Naïve | BLAST | GeneMANIA | Mashup | deepNF | NetQuilt | Graph2GO | DeepGraphGO | CFAGO | BDGO (Ours) |
|---|---|---|---|---|---|---|---|---|---|---|---|
| $\mathbf{F_{max}}$ | BPO | 0.051±0 | 0.270±0 | 0±0 | 0.075±0 | 0.394±0.006 | 0.164±0.014 | 0.335±0.01 | 0.327±0.028 | 0.439±0.007 | **0.440±0.013** |
| | MFO | 0.177±0 | 0.122±0 | 0±0 | 0.058±0 | 0.153±0.004 | 0.081±0.013 | 0.196±0.006 | 0.142±0.035 | 0.236±0.004 | **0.282±0.038** |
| | CCO | 0.121±0 | 0.196±0 | 0.031±0 | 0±0 | 0.297±0.009 | 0.138±0.013 | 0.298±0.011 | 0.209±0.023 | 0.366±0.018 | **0.421±0.013** |
| m-AUPR | BPO | 0.024±0 | 0.110±0 | 0.042±0 | 0.238±0 | 0.303±0.006 | 0.077±0.006 | 0.237±0.014 | 0.210±0.022 | 0.328±0.005 | **0.332±0.007** |
| | MFO | 0.050±0 | 0.044±0 | 0.050±0 | 0.053±0 | 0.089±0.001 | 0.045±0.007 | 0.103±0.007 | 0.080±0.021 | 0.159±0.003 | **0.172±0.014** |
| | CCO | 0.047±0 | 0.084±0 | 0.103±0 | 0.179±0 | 0.178±0.005 | 0.081±0.003 | 0.215±0.025 | 0.133±0.011 | 0.337±0.005 | **0.392±0.012** |
| M-AUPR | BPO | 0.048±0 | 0.093±0 | 0.160±0 | 0.146±0 | 0.174±0.005 | 0.081±0.004 | 0.150±0.006 | 0.133±0.008 | **0.188±0.003** | 0.171±0.004 |
| | MFO | 0.029±0 | 0.084±0 | 0.109±0 | 0.089±0 | 0.118±0.004 | 0.064±0.003 | 0.111±0.005 | 0.098±0.007 | **0.138±0.005** | 0.088±0.012 |
| | CCO | 0.060±0 | 0.082±0 | 0.150±0 | 0.104±0 | 0.155±0.009 | 0.063±0.004 | 0.159±0.021 | 0.133±0.006 | 0.210±0.007 | **0.257±0.011** |
| F1 | BPO | 0.035±0 | 0.159±0 | 0.054±0 | 0.248±0 | 0.228±0.005 | 0.114±0.017 | 0.222±0.01 | 0.238±0.012 | **0.283±0.006** | 0.264±0.007 |
| | MFO | 0.004±0 | 0.064±0 | 0.008±0 | 0.106±0 | 0.117±0.004 | 0.070±0.016 | 0.167±0.009 | 0.165±0.056 | 0.234±0.005 | **0.263±0.036** |
| | CCO | 0.070±0 | 0.107±0 | 0.123±0 | 0.202±0 | 0.205±0.009 | 0.108±0.013 | 0.261±0.015 | 0.210±0.016 | 0.314±0.007 | **0.337±0.018** |
| ACC | BPO | 0±0 | 0.071±0 | 0±0 | 0.044±0 | 0.158±0.011 | 0.048±0.007 | 0.257±0.007 | 0.153±0.034 | **0.338±0.013** | 0.331±0.012 |
| | MFO | 0±0 | 0.015±0 | 0±0 | 0.038±0 | 0.034±0.002 | 0.017±0.002 | 0.114±0.015 | 0.048±0.007 | 0.100±0.003 | **0.103±0.04** |
| | CCO | 0±0 | 0.034±0 | 0±0 | 0±0 | 0.080±0.012 | 0.037±0.005 | 0.180±0.024 | 0.066±0.011 | **0.210±0.008** | **0.210±0.041** |

The experimental results show that the performance of BDGO, CFAGO, DeepGraphGO, and Graph2GO, surpasses that of other unimodal-based methods. It indicates that multimodal data is crucial for improving the performance of protein function prediction. And owing to the pre-training (as shown in Table 9 of Appendix Section 6.5) and fine-tuning training paradigm, BDGO and CFAGO exhibit better performance. From Figure 3 (6), BDGO exhibits superior overall performance compared to CFAGO in terms of Fmax and m-AUPR. For the F1 and ACC metrics, BDGO and CFAGO show closely matched results, such as in the BPO domain, where BDGO's ACC and F1 scores differ from CFAGO by only 0.007 and 0.019, respectively. It indicates that BDGO's architecture enables a more effective learning of deep representations among multimodal features, leading to a further enhancement in overall performance. Moreover, critical difference diagrams in Figures 8, 9 and 10 of Appendix Section 6.8, further highlight BDGO's consistent advantage over other methods. In addition, we evaluate the model's ability to predict unannotated proteins in Appendix 6.4 and analyze enzyme function (EC) prediction in Appendix 6.7, demonstrating that our method outperforms CFAGO.

At the same time, we observe that BDGO does not achieve optimal results in terms of M-AUPR for BPO and MFO. This can be attributed to the fact that, in multi-label classification tasks, M-AUPR evaluates the model's predictive performance for each class individually, giving equal weight to classes with fewer samples, which may not accurately reflect the model's true performance. On the other hand, m-AUPR, which aggregates the performance across all classes, provides a more comprehensive measure of the model's overall predictive capability.

### 3.3 FEATURE EFFECTIVENESS ANALYSIS

To further evaluate the distinguishing power of the multimodal features extracted by different components of our method, Davies-Bouldin (DB)(Wu et al., 2023) scores are used. In the calculation of DB scores, GO terms are set as the labels for protein clusters, meaning proteins sharing the same GO term set are grouped into the same cluster. A lower DB score indicates that the features within clusters are more compact and that the separation between clusters is more distinct.

Based on the results in Figure 4, it is clear that the learned features of the model outperform the original input features, indicating that the components of BDGO effectively capture multimodal features. Comparing the features output by the various components of BDGO, DSM_embedding achieves the best performance across all aspects of GO. Notably, in the CCO aspect, DSM_embedding shows an improvement of at least 29.3% over other features related to CCO, demonstrating that the multi-branch dynamic feature selection mechanism better identifies features for multi-label classification. In addition, the SFE branch and IFE branch of BDGO demonstrate their respective performance ad-

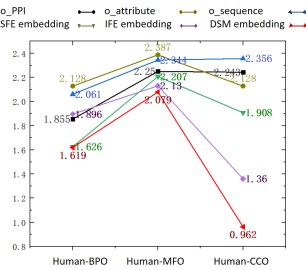

Figure 4: Davies Bouldin Score comparison of different protein feature represents. o_PPI, o_attribute and o_sequence represent the original embedding of PPI network embedding, protein attribute, and protein language model, respectively. SFE_embedding, IFE_embedding, and DSM_embedding represent the embedding from the Shared Feature Extraction branch, the embedding from the Interaction Feature Extraction branch, and the Dynamic Selection Module, respectively.

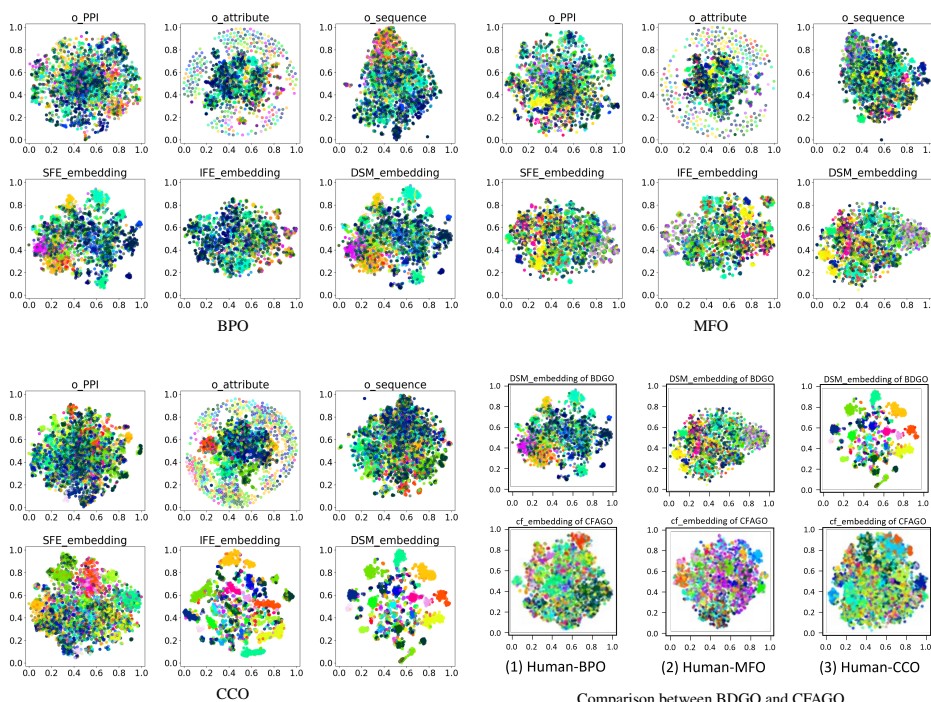

Figure 5: Visualization of different feature representations for BDGO, and comparison with CFAGO.

vantages in Figure 4, proving the necessity of integrating these two branches in the BDGO method. SFE_embedding achieves a strong score of 1.626 in the BPO aspect, suggesting that the Shared Feature Extraction contributes significantly to the model's performance in BPO.

To further analyze the discriminative power of protein features, we visualize them using t-SNE(Chatzimparmpas et al., 2020) (Figure 5). Raw input features (o_PPI, o_attribute, o_sequence) show distinct patterns but lack clear clustering boundaries. After interaction through our model, the gated features achieve a more optimal distribution. BDGO's DSM_embedding performs best, forming clearer clusters and sharper classification boundaries.

Additionally, we compare the visualization results of the final output features from BDGO and CFAGO, as shown in Figure 5 Comparison between BDGO and CFAGO. Here, DSM_embedding represents BDGO's dynamically selected features from both branches, while cf_embedding shows CFAGO's multimodal feature fusion using a multihead attention mechanism. By comparing DSM_embedding and cf_embedding, the visualization of cf_embedding shows a tendency for multi-

Table 2: Results of Ablation Studies. The overall model is denoted as 'MSLB+MILB', where 'MSLB' and 'MILB' are the backbone components: MSL-Branch and MIL-Branch. $w/o$ BInM and $w/o$ DSM represent removing the BInM and DSM modules from the overall model. $w/o$ SP-F refers to removing spatial structure features from the input, while $w/o$ SE-F indicates removing sequence features. The best results are marked in bold.

| Method | $\mathbf{F_{max}}$ | | | m-AUPR | | | M-AUPR | | | F1 | | | ACC | | |
|---|---|---|---|---|---|---|---|---|---|---|---|---|---|---|---|
| | BPO | MFO | CCO | BPO | MFO | CCO | BPO | MFO | CCO | BPO | MFO | CCO | BPO | MFO | CCO |
| MSLB | 0.428 | 0.231 | 0.388 | 0.323 | 0.153 | 0.322 | 0.154 | 0.082 | 0.188 | 0.264 | 0.240 | 0.292 | 0.324 | 0.075 | 0.168 |
| MILB | 0.396 | 0.256 | 0.377 | 0.270 | 0.112 | 0.333 | 0.156 | 0.083 | 0.229 | 0.240 | 0.142 | 0.305 | 0.313 | 0.063 | 0.210 |
| MSLB+MILB | **0.440** | **0.282** | **0.421** | **0.332** | **0.172** | **0.392** | **0.171** | 0.088 | **0.257** | **0.264** | **0.263** | **0.337** | **0.331** | 0.103 | **0.210** |
| $w/o$ BInM | 0.431 | 0.204 | 0.390 | 0.323 | 0.133 | 0.315 | 0.170 | 0.082 | 0.218 | 0.256 | 0.198 | 0.337 | 0.330 | 0.090 | 0.176 |
| $w/o$ DSM | 0.404 | 0.167 | 0.373 | 0.266 | 0.131 | 0.321 | 0.170 | 0.083 | 0.251 | 0.264 | 0.176 | 0.321 | 0.313 | 0.085 | 0.202 |
| $w/o$ SP-F | 0.216 | 0.184 | 0.265 | 0.106 | 0.102 | 0.171 | 0.104 | **0.101** | 0.112 | 0.172 | 0.174 | 0.230 | 0.152 | 0.087 | 0.156 |
| $w/o$ SE-F | 0.249 | 0.272 | 0.357 | 0.118 | 0.154 | 0.212 | 0.116 | 0.082 | 0.180 | 0.181 | 0.257 | 0.307 | 0.173 | **0.128** | 0.205 |

ple clusters to blend together compared to DSM_embedding. Particularly in the CCO aspect, BDGO demonstrates a clearer separation between different clusters, with more distinct boundaries.

## 4 ABLATION STUDIES

In this section, the contributions of each component in BDGO and the two types of features are evaluated, as shown in Table 2. Additionally, we also performed critical difference diagrams in Figures 11, 12 and 13 of Appendix 6.8, which demonstrates that each component contributes positively to the performance improvement. Further discussion on the ablation studies of the key components in the Encoder can be seen in Appendix 6.2.

**Analysis for Backbone Components.** According to lines 1,2, and 3 of Table 2, the results of the backbone network only using MSL-Branch or MIL-Branch are not as good as those using combined branches.

**Effectiveness of BInM.** Considering the correlation of features among space and sequence, this method uses the BInM block to facilitate bidirectional multimodal feature interaction before dynamic selection. As shown in the results of rows 3 and 4 in Table 2, we verify the validity of BInM for the overall model by removing it.

**Effectiveness of DSM.** To enable effective feature selection and accurate prediction of protein functions, DSM is used to adaptively select channel features most relevant to specific functional labels. At the same time, it reduces the interference and conflict caused by redundant features. As shown in rows 3 and 5 of Table 2, the dynamic selection mechanism achieved by DSM has a positive impact on protein function prediction. Furthermore, we conduct additional experiments in Table 5 in Section 6.1.2 of the Appendix, exploring different selection mechanisms of DSM, which further demonstrate its effectiveness.

**Impact of Spatial Structure and Spatial Features.** To verify the complementarity between sequence and spatial structure features, we perform an ablation study, retaining only spatial structure or sequence features. For the BInM module, it is removed as no interaction occurs with a single feature type. Rows 6 and 7 of Table 2 show that removing feature interaction significantly reduces model performance.

## 5 CONCLUSION

This method enhances the model's ability to integrate multimodal features through two key components: Bidirectional Interaction and Dynamic Selection Mechanisms. As a result, it significantly improves protein function prediction performance. Experimental results show that the BDGO method outperforms current state-of-the-art unimodal and multimodal methods across multiple metrics. These results underscore the importance of integrating multimodal data to enhance protein function prediction. It also validates the superiority of the Bidirectional Interaction Module and Dynamic Selection Module in multimodal protein data integration.

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

## 6 APPENDIX

### 6.1 MODULE EXPLORATION EXPERIMENTS

In order to further validate the rationality of the module design, the key modules in BDGO (i.e., BInM and DSM) are explored in this section.

#### 6.1.1 BINM EXPLORATION

To explore the feature interaction capability within BInM, we designed an experiment where we separately removed the cross-attention module in the BInM. BDGO-BInM-0 and BDGO-BInM-1 represent removing the bottom and top cross-attention module from BInM.

Based on the results in Table 3, we find that removing either of the cross-attention modules leads to a decline in overall performance. This validates the ability of BInM to capture interactions.

Table 3: Performance Comparison Under Different Interaction Settings. The comparison results of $F_{max}$ score, micro-averaged AUPR (m-AUPR), macro-averaged AUPR (M-AUPR), F1-score (F1), and accuracy (ACC), for BPO, MFO and CCO. The best results are highlighted in bold, and the sub-optimal results are underlined.

| Method | $\mathbf{F_{max}}$ | | | m-AUPR | | | M-AUPR | | | F1 | | | ACC | | |
|---|---|---|---|---|---|---|---|---|---|---|---|---|---|---|---|
| | BPO | MFO | CCO | BPO | MFO | CCO | BPO | MFO | CCO | BPO | MFO | CCO | BPO | MFO | CCO |
| BDGO | **0.440** | **0.282** | **0.421** | **0.332** | **0.172** | **0.392** | 0.171 | 0.088 | **0.257** | 0.264 | 0.263 | **0.337** | **0.331** | **0.103** | **0.210** |
| BDGO-BInM-0 | 0.437 | 0.262 | 0.384 | 0.313 | 0.166 | 0.314 | 0.185 | 0.079 | 0.204 | **0.272** | **0.278** | 0.329 | 0.302 | 0.051 | 0.176 |
| BDGO-BInM-1 | 0.434 | 0.227 | 0.319 | 0.308 | 0.118 | 0.215 | 0.184 | **0.093** | 0.174 | 0.269 | 0.172 | 0.251 | 0.302 | 0.092 | 0.202 |

#### 6.1.2 DSM EXPLORATION

To investigate the difference between hard gating and soft gating in the DSM module, we compare the module's performance using both gating mechanisms. Specifically, our model BDGO employs hard gating, while BDGO-Soft corresponds to the version with soft gating.

Based on the results in Table 4, we find that when soft gating is used, the overall performance of the model declines. This may be because soft gating selects features that are not decisive for the functionality.

Table 4: Performance Comparison of Hard Gating and Soft Gating in DSM Module. The comparison results of $F_{max}$ score, micro-averaged AUPR (m-AUPR), macro-averaged AUPR (M-AUPR), F1-score (F1), and accuracy (ACC), for BPO, MFO and CCO. The best results are highlighted in bold.

| Method | $\mathbf{F_{max}}$ | | | m-AUPR | | | M-AUPR | | | F1 | | | ACC | | |
|---|---|---|---|---|---|---|---|---|---|---|---|---|---|---|---|
| | BPO | MFO | CCO | BPO | MFO | CCO | BPO | MFO | CCO | BPO | MFO | CCO | BPO | MFO | CCO |
| BDGO | 0.440 | **0.282** | **0.421** | **0.332** | **0.172** | **0.392** | 0.171 | **0.088** | **0.257** | 0.264 | **0.263** | 0.337 | **0.331** | **0.103** | **0.210** |
| BDGO-Soft | **0.444** | 0.246 | 0.394 | 0.322 | 0.143 | 0.319 | **0.177** | 0.080 | 0.228 | **0.275** | 0.184 | **0.343** | 0.310 | 0.092 | 0.200 |

To investigate how many features experts should select in the DSM module for optimal performance, we conduct an additional experiment. This experiment evaluates the performance when experts in the DSM module select multiple features. Here, BDGO-DSM-$C_m^n$ represents the number of ways to select $n$ features from $m$ features without repetition. This value also determines the number of experts in the DSM module.

As shown in Table 5, BDGO achieves the best overall performance when each expert selects only a single feature.

Table 5: Performance Comparison Under Different Interaction Settings. The comparison results of $F_{max}$ score, micro-averaged AUPR (m-AUPR), macro-averaged AUPR (M-AUPR), F1-score (F1), and accuracy (ACC), for BPO, MFO and CCO. The best results are highlighted in bold, and the sub-optimal results are underlined.

| Method | $\mathbf{F_{max}}$ | | | m-AUPR | | | M-AUPR | | | F1 | | | ACC | | |
|---|---|---|---|---|---|---|---|---|---|---|---|---|---|---|---|
| | BPO | MFO | CCO | BPO | MFO | CCO | BPO | MFO | CCO | BPO | MFO | CCO | BPO | MFO | CCO |
| BDGO | **0.440** | **0.282** | **0.421** | **0.332** | **0.172** | **0.392** | 0.171 | 0.088 | **0.257** | 0.264 | **0.263** | 0.337 | **0.331** | **0.103** | 0.210 |
| BDGO-DSM-$C_6^2$ | 0.437 | 0.201 | 0.399 | 0.317 | 0.110 | 0.291 | 0.174 | **0.124** | 0.195 | **0.276** | 0.200 | 0.305 | 0.310 | 0.089 | 0.213 |
| BDGO-DSM-$C_6^3$ | 0.407 | 0.204 | 0.382 | 0.271 | 0.111 | 0.284 | 0.163 | **0.124** | 0.192 | 0.264 | 0.206 | 0.295 | 0.268 | 0.087 | **0.230** |
| BDGO-DSM-$C_6^4$ | 0.426 | 0.207 | 0.404 | 0.294 | 0.113 | 0.316 | 0.176 | 0.120 | 0.210 | 0.274 | 0.204 | **0.345** | 0.292 | 0.083 | 0.207 |
| BDGO-DSM-$C_6^5$ | 0.414 | 0.210 | 0.390 | 0.283 | 0.113 | 0.308 | **0.178** | **0.124** | 0.208 | 0.268 | 0.208 | 0.327 | 0.292 | 0.087 | 0.200 |

## 6.2 EXPLORING DIFFERENT ARCHITECTURES FOR PROTEIN FEATURE MODELING

In this work, the corresponding PSSI encoder and PSI encoder are trained according to the spatial structure feature and sequence feature in the pre-training stage. In order to verify the effectiveness of the PSSI encoder and the PSI encoder, the ablation experiment of the key module of the encoder is carried out in this section.

In this experiment, the pre-training framework is the same as stage 1 in Figure 1. Here, BiMamba Block in the pre-trained encoder can be replaced by Multihead Attention Block (Dosovitskiy et al., 2021), and vice versa. Then, we use the pre-trained encoder for feature extraction in the fine-tuning task. During fine-tuning, we only use an MLP for classification.

As shown in Table 6, the model names are formed by combining the feature and framework components with a hyphen. This indicates that the results are obtained by pre-training the feature with the framework component and then fine-tuning it.

We model the spatial structure features and sequence features using encoders with different components. According to the experimental results in Table 6, we find that the encoder using BiMamba Block as a component for modeling spatial structure features shows significant advantages in MFO and CCO. When modeling sequence features, the encoder utilizing Multihead Attention as a component demonstrates considerable advantages in BPO and CCO. Additionally, we observe that Bi-Mamba Block is significantly more efficient in processing spatial structure information, requiring much less time to predict a protein's function compared to Multihead Attention. Considering both inference efficiency and performance, we decide to use an encoder with BiMamba Block as the component for spatial structure features modeling. For sequence information, while BiMamba Block achieves better inference efficiency, we prioritize model performance and opt for Multihead Attention to model sequence features.

Table 6: Ablation experiments of key components in the pre-trained encoder. The comparison results of $F_{max}$ score, micro-averaged AUPR (m-AUPR), macro-averaged AUPR (M-AUPR), F1-score (F1), and accuracy (ACC), for BPO, MFO and CCO. The Cost Time (ms) represents the time it takes for the model to test each protein sample. The best results are highlighted in bold.

| Method | $\mathbf{F_{max}}$ | | | m-AUPR | | | Cost Time (ms) |
|---|---|---|---|---|---|---|---|
| | BPO | MFO | CCO | BPO | MFO | CCO | |
| Spatial-BiMamba Block | 0.370 | **0.240** | **0.419** | 0.244 | **0.156** | **0.371** | **0.539** |
| Spatial-Multihead Attention | **0.430** | 0.223 | 0.350 | **0.291** | 0.154 | 0.313 | 5.661 |
| Sequence-BiMamba Block | 0.290 | **0.237** | 0.308 | 0.185 | **0.157** | 0.230 | **0.280** |
| Sequence-Multihead Attention | **0.329** | 0.219 | **0.345** | **0.199** | 0.148 | **0.248** | 0.450 |

## 6.3 IMPACT OF SEQUENCE SIMILARITY ON MODEL PERFORMANCE

To ensure the validity of our experimental design and avoid potential data leakage, we analyze the sequence similarity between the test set and the combined training and validation sets for BPO, MFO, and CCO. We calculate the similarity for each of these sets and categorize the results into different similarity ranges.

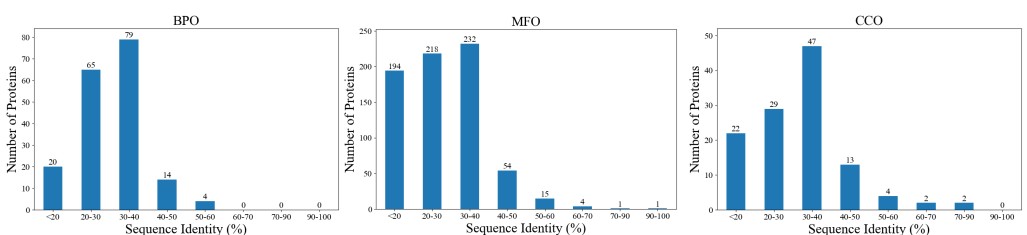

Figure 6: Distribution of sequence identity across proteins in the test dataset. The x-axis represents the sequence similarity ranges. The y-axis indicates the number of proteins within each range. Each bar denotes the number of proteins in the test set that fall within the corresponding similarity range.

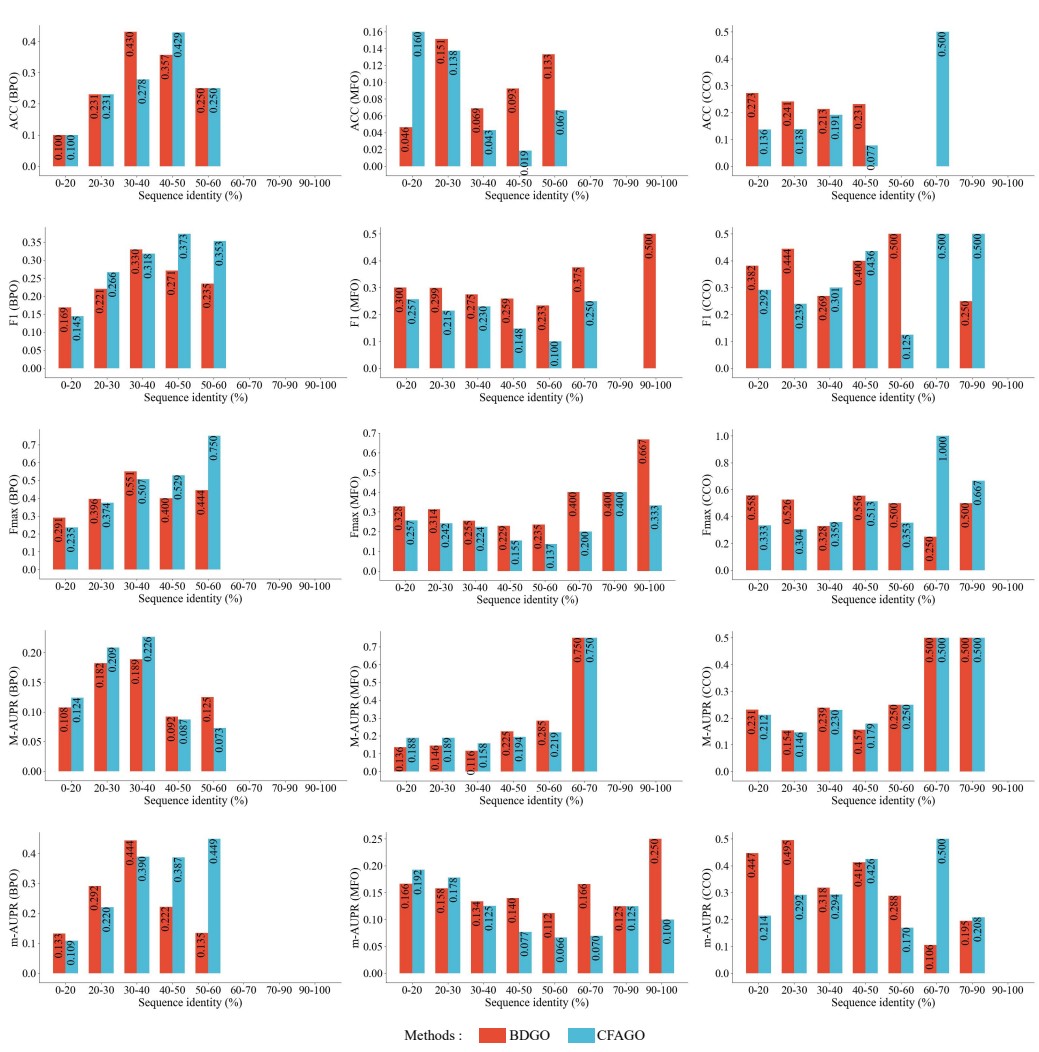

Figure 7: Performance of the different methods in Different Sequence Similarity Ranges. Figures show the results of F1-score (F1), accuracy (ACC), F-max score, macro-averaged AUPR (M-AUPR), and micro-averaged AUPR (m-AUPR) for BPO, MFO and CCO, respectively. The red pillar represents the results of the BDGO model and the blue pillar shows the results of the CFAGO model.

The following Figure 6 shows the number of proteins in the test set within each similarity range. We observe that the majority of proteins in our test set exhibit an average sequence similarity of less than 50% with the proteins in the combined training and validation sets. Only a few proteins have an average similarity greater than 70%. Based on these results, we conclude that the time-based split used in the CAFA challenge is reasonable and does not introduce significant sequence similarity overlap between our training and test sets.

As shown in Figure 3 and Table 1, the BLAST method, which relies on sequence similarity performs poorly. This further supports the notion that the sequence similarity between the test set and the combined training and validation sets is relatively low.

Additionally, we explore the model's performance in predicting protein functions within different similarity ranges. We divide the test set into 9 similarity intervals. In this figure 7, we present the performance of BDGO and CFAGO across different protein similarity ranges. For both BPO and CCO, our method demonstrates significantly better performance than CFAGO, particularly for proteins with low similarity. Specifically, range (n-m) represents the average similarity of proteins in the range of n-m. The results are shown in the following Table 7. The symbol "-" indicates that there are no proteins in that interval. Due to the scarcity of proteins with similarity above 50%, we do not provide further analysis for this group. For proteins with similarity below 50%, the model performs best in predicting proteins in the similarity range of 30 to 40 for BPO. For MFO and CCO, the model performs best for proteins in the similarity range of 0 to 30. These results show that our proposed BDGO model is stable across different sequence similarities and is not reliant on the sequence similarity of the data.

Table 7: Performance of the different methods in Different Sequence Similarity Ranges. The comparison results of $F_{max}$ score, micro-averaged AUPR (m-AUPR), macro-averaged AUPR (M-AUPR), F1-score (F1), and accuracy (ACC), for BPO, MFO and CCO. The Range represents the proteins that have average similarity within a specific range.

| Method | Range | $F_{max}$ | | | m-AUPR | | | M-AUPR | | | F1 | | | Acc | | |
|---|---|---|---|---|---|---|---|---|---|---|---|---|---|---|---|---|
| | | BPO | MFO | CCO | BPO | MFO | CCO | BPO | MFO | CCO | BPO | MFO | CCO | BPO | MFO | CCO |
| | 0-100 | 0.439 | 0.236 | 0.366 | 0.328 | 0.159 | 0.337 | 0.188 | 0.138 | 0.210 | 0.283 | 0.234 | 0.314 | 0.338 | 0.100 | 0.210 |
| | 0-20 | 0.235 | 0.257 | 0.333 | 0.109 | 0.192 | 0.214 | 0.124 | 0.188 | 0.212 | 0.145 | 0.257 | 0.292 | 0.100 | 0.160 | 0.136 |
| | 20-30 | 0.374 | 0.242 | 0.304 | 0.220 | 0.178 | 0.292 | 0.209 | 0.189 | 0.146 | 0.266 | 0.215 | 0.239 | 0.231 | 0.138 | 0.138 |
| | 30-40 | 0.507 | 0.224 | 0.359 | 0.390 | 0.125 | 0.294 | 0.226 | 0.158 | 0.230 | 0.318 | 0.230 | 0.301 | 0.278 | 0.043 | 0.191 |
| CFAGO | 40-50 | 0.529 | 0.155 | 0.513 | 0.387 | 0.077 | 0.426 | 0.087 | 0.194 | 0.179 | 0.373 | 0.148 | 0.436 | 0.429 | 0.019 | 0.077 |
| | 50-60 | 0.750 | 0.137 | 0.353 | 0.449 | 0.066 | 0.170 | 0.073 | 0.219 | 0.250 | 0.353 | 0.100 | 0.125 | 0.250 | 0.067 | 0.000 |
| | 60-70 | - | 0.200 | 1.000 | - | 0.070 | 0.500 | - | 0.750 | 0.500 | - | 0.250 | 0.500 | - | 0.000 | 0.500 |
| | 70-90 | - | 0.400 | 0.667 | - | 0.125 | 0.208 | - | 0.000 | 0.500 | - | 0.000 | 0.500 | - | 0.000 | 0.000 |
| | 90-100 | - | 0.333 | - | - | 0.100 | - | - | 0.000 | - | - | 0.000 | - | - | 0.000 | - |
| | 0-100 | 0.440 | 0.282 | 0.421 | 0.332 | 0.172 | 0.392 | 0.171 | 0.088 | 0.257 | 0.264 | 0.263 | 0.337 | 0.331 | 0.103 | 0.210 |
| | 0-20 | 0.291 | 0.328 | 0.558 | 0.133 | 0.166 | 0.447 | 0.108 | 0.136 | 0.231 | 0.169 | 0.300 | 0.382 | 0.100 | 0.046 | 0.273 |
| | 20-30 | 0.396 | 0.314 | 0.526 | 0.292 | 0.158 | 0.495 | 0.182 | 0.146 | 0.154 | 0.221 | 0.299 | 0.444 | 0.231 | 0.151 | 0.241 |
| | 30-40 | 0.551 | 0.255 | 0.328 | 0.444 | 0.134 | 0.318 | 0.189 | 0.116 | 0.239 | 0.330 | 0.275 | 0.269 | 0.430 | 0.069 | 0.213 |
| BDGO (Ours) | 40-50 | 0.400 | 0.229 | 0.556 | 0.222 | 0.140 | 0.414 | 0.092 | 0.225 | 0.157 | 0.271 | 0.259 | 0.400 | 0.357 | 0.093 | 0.231 |
| | 50-60 | 0.444 | 0.235 | 0.500 | 0.135 | 0.112 | 0.288 | 0.125 | 0.285 | 0.250 | 0.235 | 0.233 | 0.500 | 0.250 | 0.133 | 0.000 |
| | 60-70 | - | 0.400 | 0.250 | - | 0.166 | 0.106 | - | 0.750 | 0.500 | - | 0.375 | 0.000 | - | 0.000 | 0.000 |
| | 70-90 | - | 0.400 | 0.500 | - | 0.125 | 0.195 | - | 0.000 | 0.500 | - | 0.000 | 0.250 | - | 0.000 | 0.000 |
| | 90-100 | - | 0.667 | - | - | 0.250 | - | - | 0.000 | - | - | 0.500 | - | - | 0.000 | - |

## 6.4 EVALUATING MODEL PREDICTIONS ON UNANNOTATED PROTEINS

To evaluate the reliability of BDGO in predicting the functions of unannotated proteins, we design an additional experiment. We download 272 unverified human protein records from the UniProt database. After filtering out proteins lacking PPI data, we obtain a total of 136 protein samples. The test results, shown in Table 8, indicate that our model performs well in terms of accuracy. This is primarily because most of the labels for these 136 proteins do not fall within the predefined sets of 45 labels (BPO), 38 labels (MFO), and 35 labels (CCO), resulting in a majority of zero labels.

Table 8: Performance of BDGO on Unannotated Proteins. The comparison results of $F_{max}$ score, micro-averaged AUPR (m-AUPR), macro-averaged AUPR (M-AUPR), F1-score (F1), and accuracy (ACC), for BPO, MFO and CCO on the annotated and unannotated datasets.

| Method | $\mathbf{F_{max}}$ | | | m-AUPR | | | M-AUPR | | | F1 | | | ACC | | |
|---|---|---|---|---|---|---|---|---|---|---|---|---|---|---|---|
| | BPO | MFO | CCO | BPO | MFO | CCO | BPO | MFO | CCO | BPO | MFO | CCO | BPO | MFO | CCO |
| BDGO (annotated) | 0.440 | 0.282 | 0.421 | 0.332 | 0.172 | 0.392 | 0.171 | 0.088 | 0.257 | 0.264 | 0.263 | 0.337 | 0.331 | 0.103 | 0.210 |
| BDGO (unannotated) | 0.143 | 0.025 | 0.232 | 0.015 | 0.010 | 0.101 | 0.084 | 0.041 | 0.164 | 0.024 | 0.022 | 0.115 | 0.596 | 0.559 | 0.449 |

## 6.5 ABLATION STUDY ON THE EFFECTIVENESS OF PRE-TRAINING

To evaluate the effectiveness of BDGO pre-training, we design a supplementary experiment. We compare the model's performance with and without pre-training using an ablation study. Specifically, we use the same network structure and assess the fine-tuned results. As shown in Table 9, the model with BDGO pre-training consistently outperforms the one without, highlighting the importance of pre-training in improving model performance.

Table 9: Ablation Study Results on Pre-training Effectiveness. The comparison results of $F_{max}$ score, micro-averaged AUPR (m-AUPR), macro-averaged AUPR (M-AUPR), F1-score (F1), and accuracy (ACC), for BPO, MFO and CCO. The best results are highlighted in bold.

| Method | $\mathbf{F_{max}}$ | | | m-AUPR | | | M-AUPR | | | F1 | | | ACC | | |
|---|---|---|---|---|---|---|---|---|---|---|---|---|---|---|---|
| | BPO | MFO | CCO | BPO | MFO | CCO | BPO | MFO | CCO | BPO | MFO | CCO | BPO | MFO | CCO |
| BDGO | **0.440** | **0.282** | **0.421** | **0.332** | **0.172** | **0.392** | **0.171** | 0.088 | **0.257** | **0.264** | **0.263** | **0.337** | 0.331 | **0.103** | 0.210 |
| BDGO $w/o$ pretrain | 0.389 | 0.176 | 0.386 | 0.195 | 0.071 | 0.225 | 0.135 | **0.105** | 0.168 | 0.248 | 0.153 | 0.297 | **0.335** | 0.144 | **0.269** |

## 6.6 COMPARISON WITH STRUCTURE-BASED AND PLM-BASED METHODS

To ensure a fair comparison, we conduct an additional experiment, evaluating the performance of the structure-based method DeepFRI, and PredGO method using a protein language model (PLM). The results demonstrate that our approach outperforms both methods in the BPO and CCO aspects.

Table 10: Comparison with Structure-based and PLM-based methods. The comparison results of $F_{max}$ score, and micro-averaged AUPR (m-AUPR) for BPO, MFO and CCO. The best results are highlighted in bold.

| Method | Focus | $\mathbf{F_{max}}$ | | | m-AUPR | | |
|---|---|---|---|---|---|---|---|
| | | BPO | MFO | CCO | BPO | MFO | CCO |
| DeepFRI | Structure based | 0.362 | **0.461** | 0.385 | 0.308 | **0.382** | 0.360 |
| PredGO | Structure + PLM based | 0.108 | 0.455 | 0.252 | 0.058 | 0.254 | 0.183 |
| BDGO (Ours) | Multi-modal based | **0.440** | 0.282 | **0.421** | **0.332** | 0.172 | **0.392** |

## 6.7 ENZYME FUNCTION PREDICTION USING BDGO

In this subsection, we evaluate the performance of BDGO on enzyme function (EC) prediction. Following the experimental setup of DeepFRI (Gligorijević et al., 2021) and GraphEC (Song et al., 2024), we construct an EC dataset for this task. The dataset consists of 2,331 proteins in the training set, 364 proteins in the validation set, and 290 proteins in the test set, with a total of 1,130 unique labels. The hyperparameters of the CFAGO model are configured as described in its original paper. For BDGO, the learning rate is set to 1e-3 , while other settings remain consistent with those used in this study.

The experimental results, presented in Table 11, show that BDGO achieves superior overall performance compared to the baselines. This indicates that the structure of BDGO is effective for EC prediction in the classification task.

Table 11: Performance Comparison of BDGO and Baseline Methods on EC Prediction. The comparison results of $F_{max}$ score, micro-averaged AUPR (m-AUPR), macro-averaged AUPR (M-AUPR), F1-score (F1), and accuracy (ACC). The best results are highlighted in bold.

| Method | Fmax | m-AUPR | M-AUPR | F1 | ACC |
|---|---|---|---|---|---|
| CFAGO | 0.452 | 0.420 | 0.117 | 0.265 | 0.407 |
| BDGO (Ours) | **0.831** | **0.834** | **0.120** | **0.432** | **0.766** |

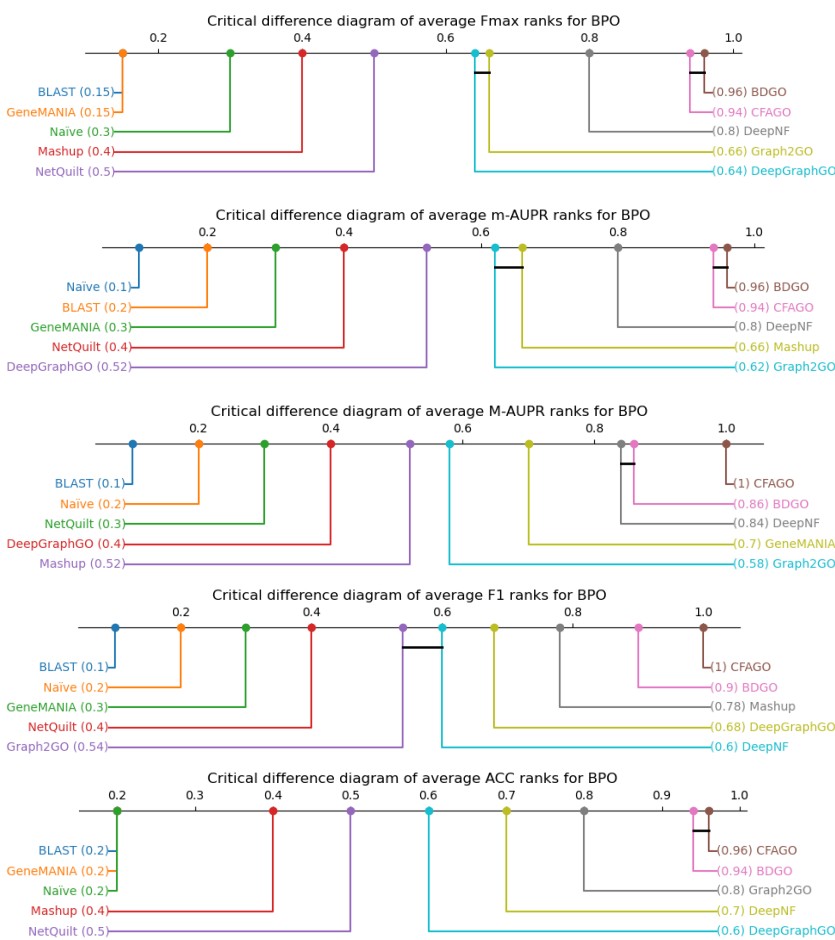

Figure 8: Critical Difference Diagram for Comparative Experiments on BPO

## 6.8 CRITICAL DIFFERENCE DIAGRAMS FOR STATISTICAL COMPARISON

To assess the significance of the results and compare the performance of different approaches, we use critical difference diagrams. These diagrams, as described in scikit-posthocs documentation, are particularly useful in visualizing whether differences between approaches are statistically significant.

The critical difference diagrams used for the comparative experiments are shown in Figures 8, 9 and 10.The critical difference diagrams used for the module ablation experiments are shown in Figures 11, 12 and 13.

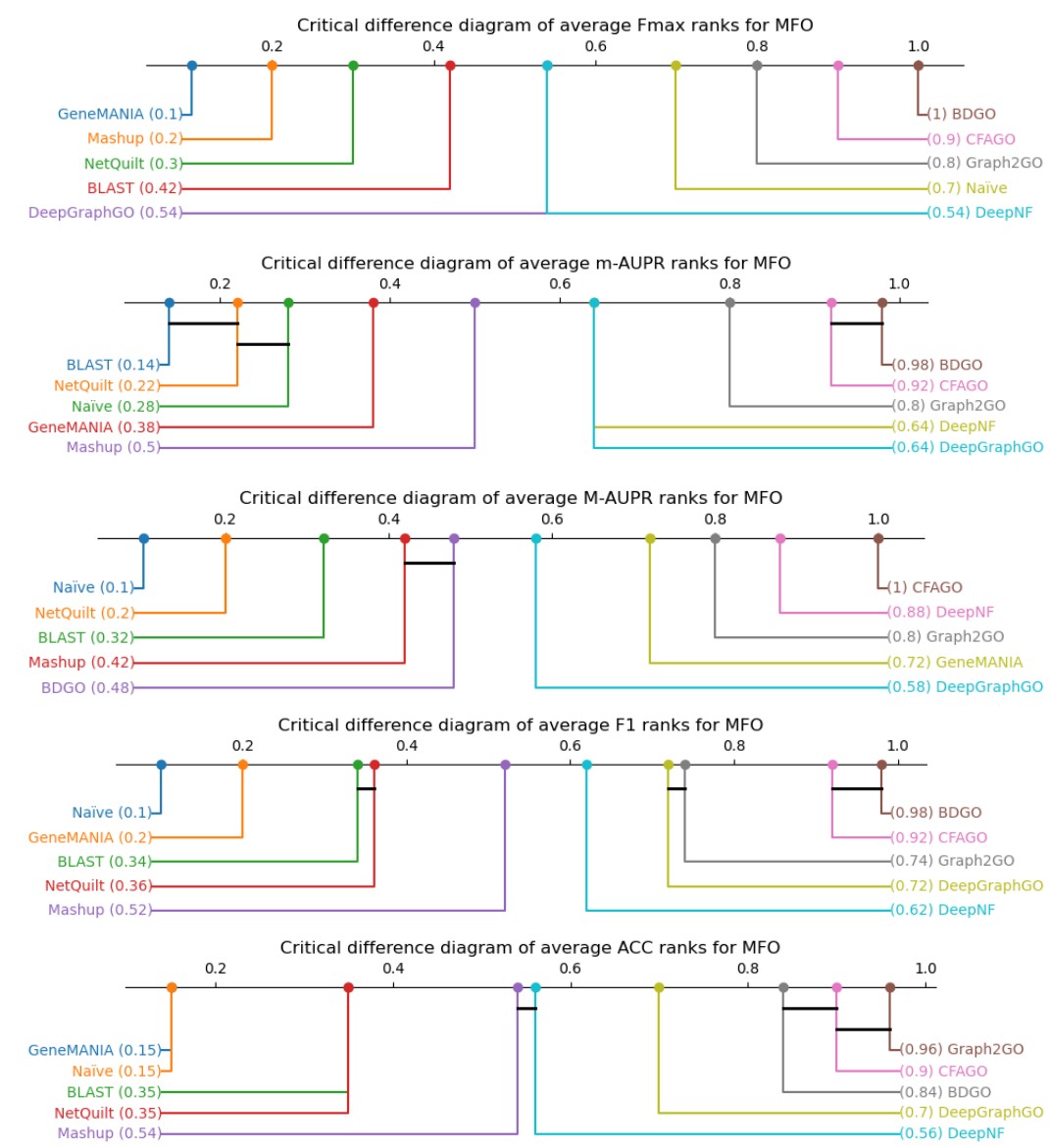

Figure 9: Critical Difference Diagram for Comparative Experiments on MFO

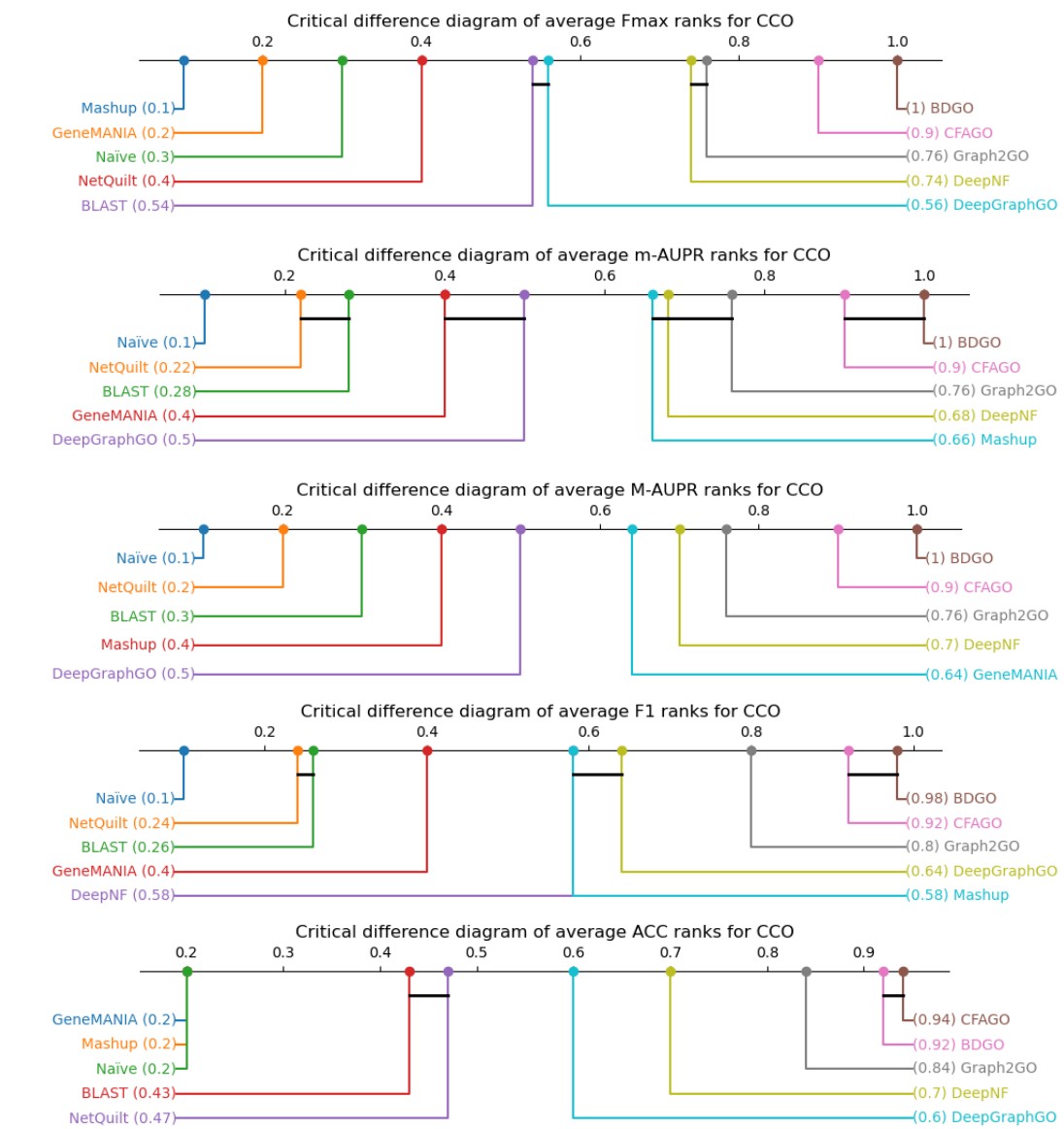

Figure 10: Critical Difference Diagram for Comparative Experiments on CCO

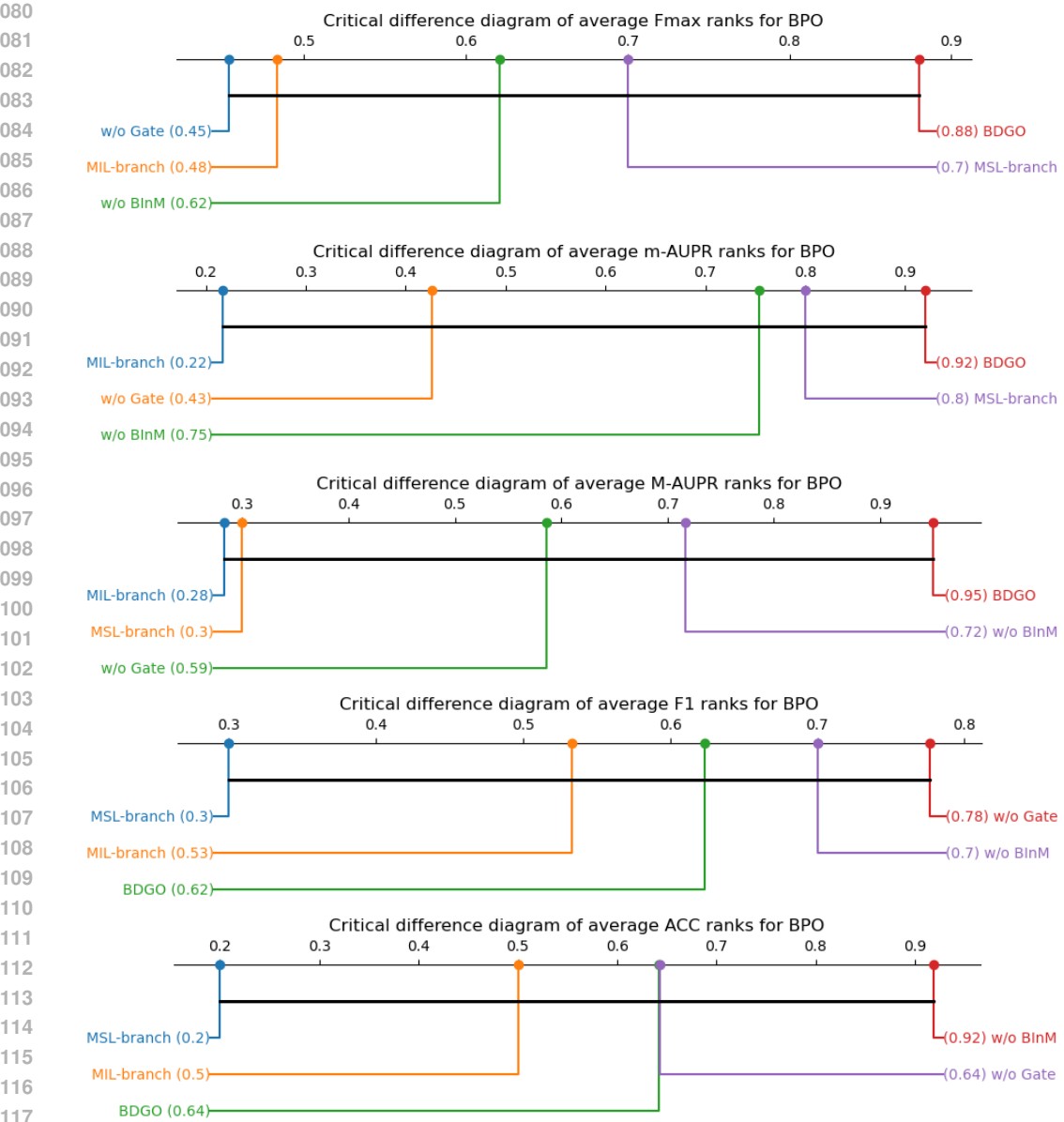

Figure 11: Critical Difference Diagram for Module Ablation on BPO

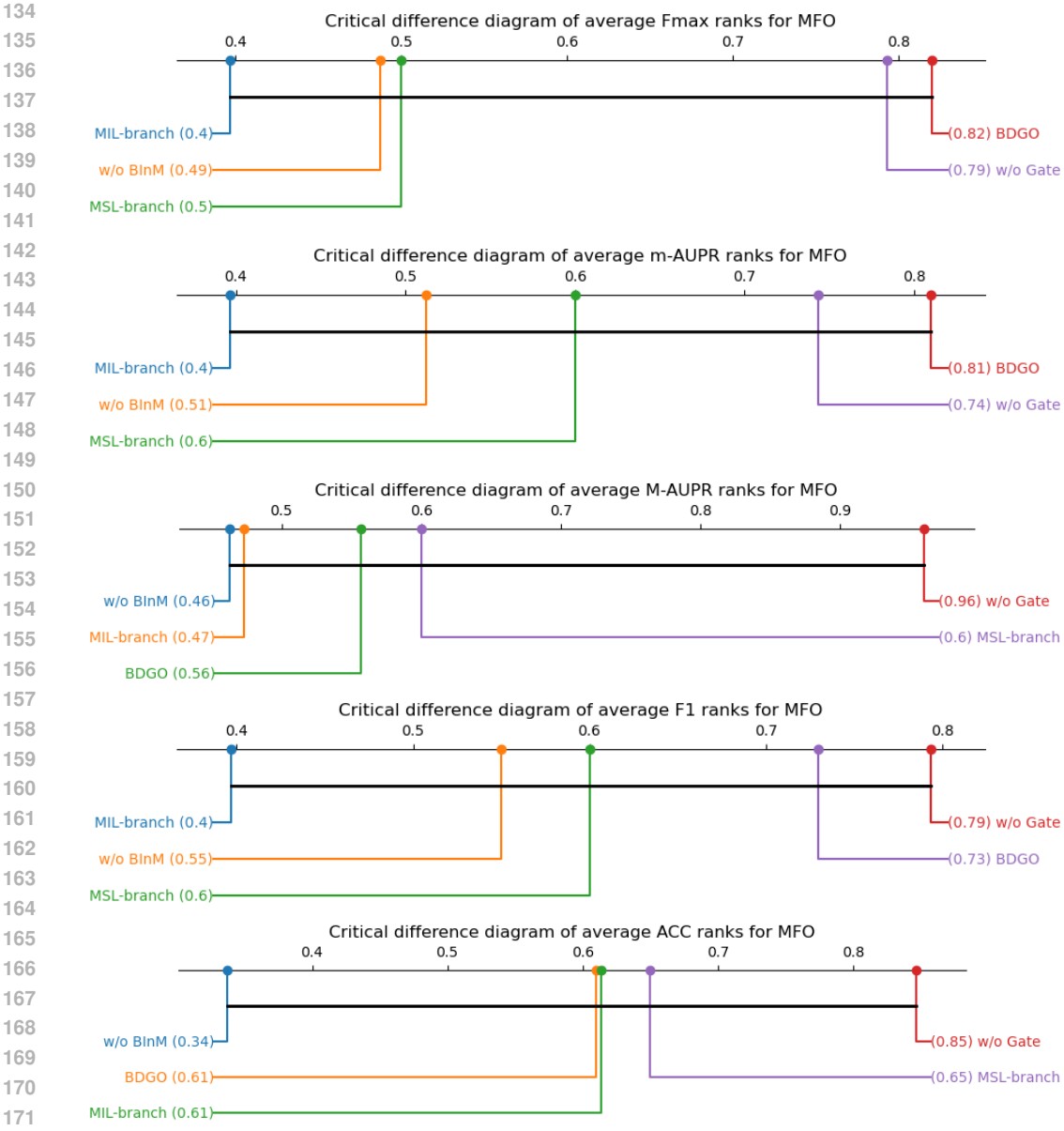

Figure 12: Critical Difference Diagram for Module Ablation on MFO

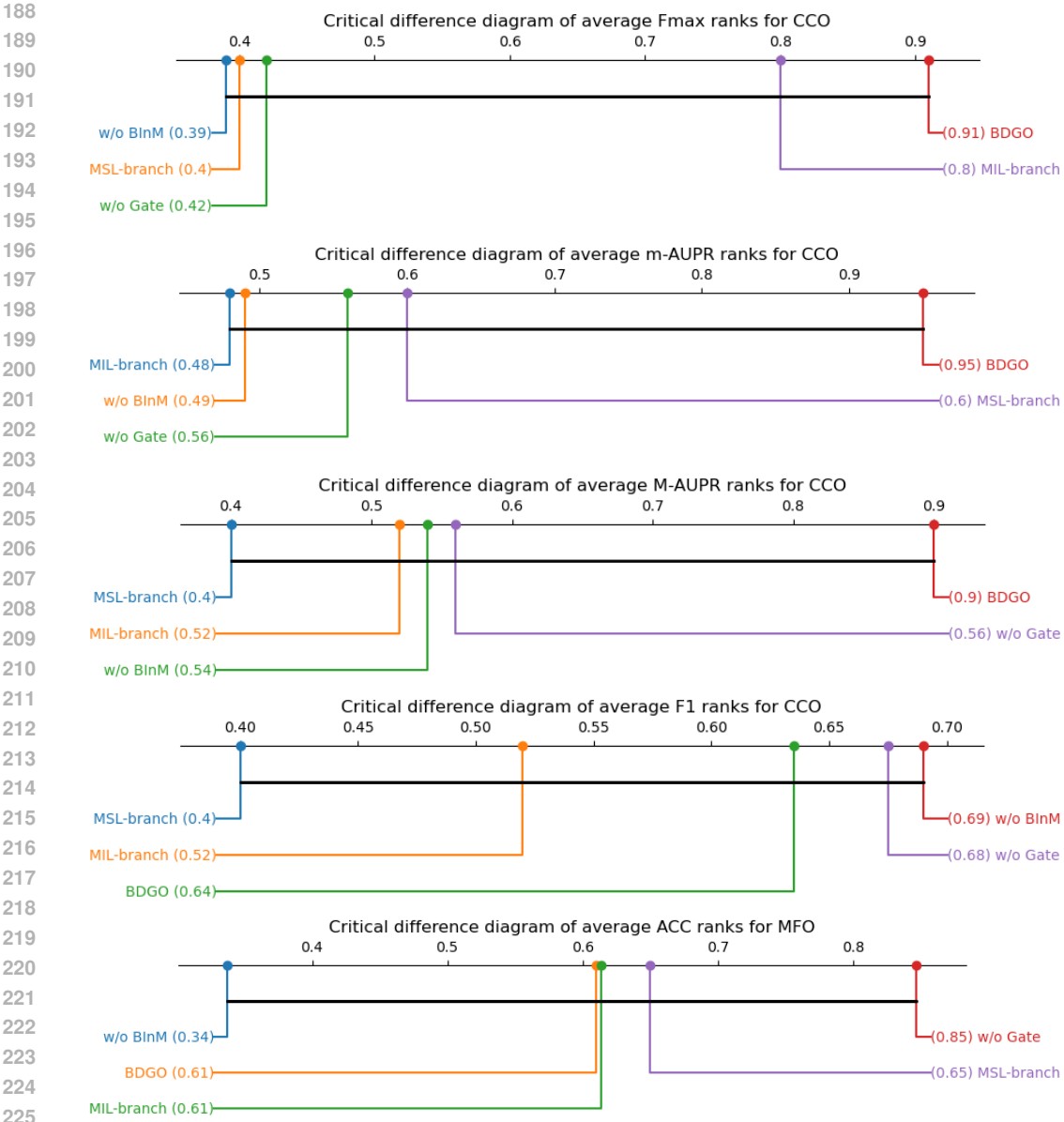

Figure 13: Critical Difference Diagram for Module Ablation on CCO

