# OpenReview forum: "Improving Multimodal Protein Function Prediction Using Bidirectional Interaction and Dynamic Selection Mechanisms"
_ICLR.cc/2025/Conference — ICLR 2025 Conference Withdrawn Submission_

### Official Review · Reviewer_2RUc · 2024-10-29

**Soundness:** 1
**Presentation:** 2
**Contribution:** 1
**Rating:** 3
**Confidence:** 4

**Summary:**

This paper introduced BDGO a framework for protein function prediction. It combines Mamba architecture and a interaction module for ontology prediction with pre-training on both the sequence and spatial modules. The authors show marginal performance improvement over some existing sequence based and PPI(protein-protein interaction) network based methods for protein function prediction.

**Strengths:**

1. The authors demonstrated the Mamba architecture combined with protein language model can improve prediction performance on existing sequence based and PPI based methods such as BLAST[1] and interaction network based methods such as Mashup[2] and NetQuilt[3].
2. Ablation studies showed the importance of the protein language models(ProtT5)[4] as a featurization tool.

**Weaknesses:**

1. Lack of motivation for the proposed architecture. The authors proposed the used of BiMamba as extension of the Mamba[5] architecture. However, no specific motivation was given on how the author hypothesize such architecture’s advantage on modeling protein sequences. very generic terms were used to justified such model selection. ‘’*BiMamba introduces a novel bidirectional selective scanning mechanism designed for protein information, which can take into account both the information at the beginning
and end of spatial features. This design allows BiMamba to capture details and context information in the spatial features of proteins.*” How does BOW encoding of domain and localization benefits from your “Spatial” encoding?
2. In the benchmarking effort, the authors only compared with methods that used naive sequence input and PPI network information. However, the proposed method uses protein language model to featurize the input sequence and it’s widely accepted in the community that the use of protein language model and significantly improve model performance[6]. The lack of acknowledgement in the benchmarking effort either shows the authors’ lack of knowledge in the field of protein function prediction or potentially intentional misleading benchmarking setup.
3. The failure to extend the framework to enzyme function(EC) prediction. It has been the standard for protein function prediction framework to include enzyme function prediction as part of evaluation. The authors failed to include such results.
4. The authors directly borrowed a dataset from a previous study CFAGO[7] which is a time stamp based split. For machine learning methods, it is very important to check for sequence similarity between the training and testing set to avoid leakage and inflated results.
5. Overall, this study showed a arbitrary deep learning architecture applied to protein function prediction. The motivation and hypothesis is not well justified for use of such model and the benchmarking effort is lacking in both comprehensiveness and rigor.

**Questions:**

1. What specific aspect of the proposed model contributed to the performance improvement? The use of ProtT5 against CFAGO is not a fair comparison because  CFAGO does not use protein language model as part of the featurization pipeline.
2. How does a BOW encoding PPI matrix utilize the space state model proposed ?
3. Why not include results for EC number prediction in your results?
4. Why not also benchmark using a sequence similarity based split?

[1]Altschul, Stephen F., et al. "Basic local alignment search tool." *Journal of molecular biology* 215.3 (1990): 403-410.
[2]Cho, Hyunghoon, Bonnie Berger, and Jian Peng. "Compact integration of multi-network topology for functional analysis of genes." *Cell systems* 3.6 (2016): 540-548.
[3]Barot, Meet, et al. "NetQuilt: deep multispecies network-based protein function prediction using homology-informed network similarity." *Bioinformatics* 37.16 (2021): 2414-2422.
[4]Elnaggar, Ahmed, et al. "Prottrans: Toward understanding the language of life through self-supervised learning." *IEEE transactions on pattern analysis and machine intelligence* 44.10 (2021): 7112-7127.
[5]Gu, Albert, and Tri Dao. "Mamba: Linear-time sequence modeling with selective state spaces." *arXiv preprint arXiv:2312.00752* (2023).
[6] Kulmanov, Maxat, et al. "Protein function prediction as approximate semantic entailment." *Nature Machine Intelligence* 6.2 (2024): 220-228.
[7]Wu, Zhourun, et al. "CFAGO: cross-fusion of network and attributes based on attention mechanism for protein function prediction." *Bioinformatics* 39.3 (2023): btad123.

---

> ### Author Response · Authors · 2024-11-24
> **Response to Reviewer 2RUc (Part 1/2)**
>
> Thank you very much for your comments. We respond to your concerns as follows. The modified parts in the revised paper are highlighted in blue.
>
> ------
>
> **W1.** Thanks for your suggestions. In this work, we did not use the BiMamba Block as a component of the Encoder to model the protein sequence, but used the BiMamba Block as a component of the Encoder to model the spatial features of the protein (PPI, subcellular localization domain). To further verify the effectiveness of BiMamba Block in Encoder, we have added an experiment in **Table 6 of Appendix Section 6.2**. In this experiment, the pre-trained framework is the same as stage 1 in the paper. In pre-trained encoder, BiMamba Block can be replaced by Multihead Attention Block, and vice versa. Similarly, we take the pre-trained encoder and use it for feature extraction for the fine-tuning task. In fine tuning, we use only one multi-layer perceptron for classification. The experimental results can be seen as follows:
>
> | **Method**                   | $F\_{max}$ |            |            | **m-AUPR** |            |            |
> | :--------------------------- | :--------: | :--------: | :--------: | :--------: | :--------: | :--------: |
> |                              |  **BPO**   |  **MFO**   |  **CCO**   |  **BPO**   |  **MFO**   |  **CCO**   |
> | Spatial-BiMamba  Block       |   0\.370   | **0\.240** | **0\.419** |   0\.244   |   0\.156   | **0\.371** |
> | Spatial-Multihead Attention  | **0\.430** |   0\.223   |   0\.350   | **0\.291** |   0\.154   |   0\.313   |
> | Sequence-BiMamba  Block      |   0\.290   |   0\.237   |   0\.308   |   0\.185   | **0\.157** |   0\.230   |
> | Sequence-Multihead Attention |   0\.329   |   0\.219   |   0\.345   |   0\.199   |   0\.148   |   0\.248   |
>
>
> As shown in experimental results, we find that the approach using BiMamba Block as a component for modeling spatial features shows significant advantages in MFO and CCO. When modeling sequence features, the method utilizing Multihead Attention as a component demonstrates considerable advantages in BPO and CCO. Furthermore, since the two types of features exhibit complementarity in different aspects, we choose to use BiMamba Block for modeling spatial features and MultiheadAttention for modeling sequence features.
>
>  As for the BOW encoding, in this paper, we do not propose a new coding approach, but classify domain and subcellular location and PPI as spatial information.
>
> ------
>
> **W2.** Thanks to reviewer for the critical observations of our benchmark settings. We acknowledge that protein language modeling has indeed become a powerful tool for protein function prediction, and *many protein function prediction methods also use protein language modeling as feature extractors, such as GAT-GO[2] using protein structural characteristics and SPROF-GO[4] based on protein sequence. However, we note that SPROF-GO, GAT-GO and other papers do not compare with methods using protein language models. At the same time, our focus is on the interaction of protein features and the effective capture of relevant features (BInM and DSM), so this aspect was not considered in the original paper.
>
> In addition, we pay high attention to the comments of reviewer, and show through the above supplementary experiments (i.e., **rows 6 and 7 of Table 2 in Section 4: Ablation Studies**) that sequence features extracted from protein language models contribute to improving model performance, but are not a decisive factor. Because in the feature ablation experiments, we found that models with interaction between spatial and sequence features clearly outperform models that only use sequence features extracted by large models.
>
> ------
>
> **Q1.** We sincerely thank the reviewers for their valuable questions. Our proposed model includes two key components: the BInM and DSM modules. We believe both modules contribute significantly to the performance improvements of the model. This can be observed in **Table 2** of the Ablation Studies, where removing either the BInM or DSM module results in a substantial performance drop. Additionally, our pretraining phase also plays a critical role in enhancing the model's performance, as detailed in **Appendix Section 6.5**.
>
>  As for the unfair comparison. We also appreciate the reviewers’ concerns regarding this issue. As clarified in the response to **Weakness 2**, we would like to address the following points: 1) Protein language models used as a feature extraction pipeline do not exhibit overwhelming superiority in performance when compared with spatial features. Therefore, we have not intentionally biased the benchmark comparisons. 2) Employing protein language models for characterizing protein sequences is a common practice. The approaches like SPROF-GO[4] and GAT-GO[2] also utilize protein language models without necessarily comparing them against methods that do not use these models.
>
> ------

---

> > ### Author Response · Authors · 2024-11-24
> > **Response to Reviewer 2RUc (Part 2/2)**
> >
> > **Q2.** We sincerely thank the reviewer for the insightful questions. The adjacency matrix derived from the PPI network, as well as the Bag-of-Words (BOW) encoding of protein subcellular localization and domain information, are collectively considered as spatial features. These spatial features can be represented as $X\in R^{2\times D}$ , where $D$ denotes the feature dimension. Subsequently, $X$ is passed through an MLP for dimensional alignment, after which the aligned $X$ is fed into a series of stacked BiMamba Blocks. These modules are specifically designed to model the spatial features. Finally, the output of this process serves as the feature representation used for downstream tasks.
> >
> > ------
> >
> > **W3/Q3.** Thank you for your comments. However, according to our observation, recent methods such as FFPred-GAN [5], DeepSS2GO [3], PO2GO [1] and etc., do not include enzyme function prediction as part of the evaluation. Our experimental design was primarily based on the methodology of the comparison methods, which is why we did not consider EC prediction in our study. However, we appreciate the reviewer’s suggestion and have found a relevant paper DeepFRI[6] that includes EC prediction. We will actively pursue this line of research in future work.
> >
> > ------
> >
> > **W4/Q4.** Thank you for the effective comments. In order to further verify the sequence similarity between the test set and the training set and the protein in the validation set, we calculated the similarity between the test set and the corresponding training set and the validation set of the BPO, MFO and CCO, respectively. Finally, we counted the number of proteins in different similarity intervals in the test set. The results can be seen in **Table 7 and Figure 6 of Appendix Section 6.3**. The X-axis represents the sequence similarity interval, the Y-axis represents the number of proteins, and each column marks the number of proteins in the corresponding similarity interval in the test set.
> >
> > We found that most of the test set proteins in our data set had an average similarity of less than 50% to the training set and validation set combined proteins, with only a few proteins achieving an average similarity of 70%. Therefore, we believe that it is reasonable to follow the time node partitioning method in CAFA competition, and it is no longer necessary to divide data sets according to sequence similarity. In addition, in our comparison experiment, BLAST method is a method based on sequence similarity, but its performance is poor, which also indicates that the sequence similarity between our test set and the training set and the verification set is not high.
> >
> > As shown in **Table 7**, due to the scarcity of proteins with similarity above 50%, we do not provide further analysis for this group. For proteins with similarity below 50%, the model performs best in predicting proteins in the similarity range of 30 to 40 for BPO. For MFO and CCO, the model performs best for proteins in the similarity range of 0 to 30.
> >
> >
> >
> > [1] Li, Wenjing, et al. "Partial order relation–based gene ontology embedding improves protein function prediction." *Briefings in Bioinformatics* 25.2 (2024): bbae077.
> >
> > [2] Lai, Boqiao, et al. "Accurate protein function prediction via graph attention networks with predicted structure information." *Briefings in Bioinformatics* 23.1 (2022): bbab502.
> >
> > [3] Song, Fu V., et al. "DeepSS2GO: protein function prediction from secondary structure." *Briefings in Bioinformatics* 25.3 (2024): bbae196.
> >
> > [4] Yuan, Qianmu, et al. "Fast and accurate protein function prediction from sequence through pretrained language model and homology-based label diffusion." *Briefings in bioinformatics* 24.3 (2023): bbad117.
> >
> > [5] Wan, Cen, et al. "Protein function prediction is improved by creating synthetic feature samples with generative adversarial networks." *Nature Machine Intelligence* 2.9 (2020): 540-550.
> >
> > [6] Gligorijević, Vladimir, et al. "Structure-based protein function prediction using graph convolutional networks." *Nature communications* 12.1 (2021): 3168.

---

> > > ### Comment · Reviewer_2RUc · 2024-11-25
> > >
> > > Q2 -> I don't have a problem with how the features are encoded. My issue lies with the authors' framing of their architecture. The proposed model doesn't directly encode the spatial features. Calling it a spatial encoder is misleading. You won't call an SVM that operates on k-mer encoding of a sequence a "sequential model," right? This response is insufficient in addressing my prior concern.
> > >
> > > Q3 -> Again, "We are not doing it because some other papers didn't do it." In contrast, the majority of recent papers on the same object adopted EC number prediction as a standard benchmark. This comment didn't address my original concern.
> > >
> > > W4/Q4 -> Table 7 is probably the most bazaar table I have seen so far. Sequence similarity split on a specific range of sequence similarity? Please do all your audience a favor and use the standard sequence similarity split that includes EVERYTHING below a certain similarity. Also, you should compare your method with OTHERS on different thresholds to demonstrate better generalizability. This table didn't address my concern at all.

---

> > > > ### Author Response · Authors · 2024-11-29
> > > > **The Second Round of Responses to Reviewer 2RUc (Part 3/3)**
> > > >
> > > > **Q2.** Thank the reviewer for the kind reminder. We have renamed the Spatial Encoder in this paper. We have renamed it the **Protein Spatial Structure Information (PSSI) Encoder**. The PSSI Encoder can effectively map high-dimensional input data into a low-dimensional latent space. Similarly, we have renamed the Sequence Encoder to the **Protein Sequence Information (PSI) Encoder**. The Decoder has also been renamed accordingly.
> > > >
> > > > ------
> > > >
> > > > **Q3.** Thanks for the comments. According to the reviewer's suggestion, we have added the experiment of enzyme function (EC) prediction. We have evaluated the performance of BDGO on EC number prediction in **Appendix 6.7**. The experimental results show that BDGO achieves superior overall performance compared to CFAGO [5], which **indicates that the structure of BDGO is effective for EC number prediction.** The experimental results can be seen as follows:
> > > >
> > > > | Method      |    $F_{max}$    |   m-AUPR   |   M-AUPR   |     F1     |    ACC     |
> > > > | :---------- | :--------: | :--------: | :--------: | :--------: | :--------: |
> > > > | CFAGO [5]      |   0\.452   |   0\.420   |   0\.117   |   0\.265   |   0\.407   |
> > > > | BDGO (Ours) | **0\.831** | **0\.834** | **0\.120** | **0\.432** | **0\.766** |
> > > >
> > > >
> > > >
> > > > [5] Zhourun Wu, et al. “Cfago: cross-fusion of network and attributes based on attention mechanism for protein function prediction”. *Bioinformatics*, 39(3):btad123, 2023.
> > > >
> > > > ------
> > > >
> > > > **W4/Q4.** Thanks for your suggestions. We used Diamond to calculate sequence similarity and referred to HiFun [6] for the classification of sequence similarity ranges. Based on this, we added a comparison between BDGO and CFAGO. Additionally, due to the large amount of tabular data that is difficult to analyze, we have presented the data in the form of bar charts in **Appendix 6.3**, **Figure 7**. For proteins with similarity below 50%, BDGO performs best in predicting proteins in the similarity range of 30 to 40 for BPO. For MFO and CCO, BDGO performs best for proteins in the similarity range of 0 to 30. Compared to CFAGO, **we find that BDGO is overall more stable across different sequence similarities and less dependent on the sequence similarity of the data.**
> > > >
> > > >
> > > >
> > > > [6] Jun Wu, et al., "HiFun: homology independent protein function prediction by a novel protein-language self-attention model", *Briefings in Bioinformatics*, 24(5): bbad311, 2023.

---

> > ### Comment · Reviewer_2RUc · 2024-11-24
> >
> > W1 -> Mamba was proposed to extend the context window while managing the compute requirement compared to the transformer construction. It's not dominantly superior to multihead attention, as shown in your new benchmarks. I don't see a reason to use it here other than, "There is a new model! Let's use it on whatever we are working on." Your answer is inefficient. Also, please be clear and do not imply that the Mamba architecture proposed directly encodes spatial features. Instead, it's just encoding a BOW representation of spatially derived features.
> >
> > W2 -> I found it interesting that the authors insist on comparing with methods that don't use the same set of features and claim the results are still somehow fair. I don't agree with this argument, and this response did not address my concerns at all. "We are not doing sufficient benchmarking because others didn't do it" is not responsible research practice. I will urge the authors to reconsider their approach to conduct responsible research that benefits the scientific community instead of their publication needs.
> >
> > Q2 -> "1) Protein language models used as a feature extraction pipeline do not exhibit overwhelming superiority in performance when compared with spatial features. Therefore, we have not intentionally biased the benchmark comparisons." This is simply a false statement. If the authors didn't intend to mislead their audience, I would suggest the authors be more careful with the recent literature. This is not a sufficient response to address my concern.

---

> ### Author Response · Authors · 2024-11-29
> **The Second Round of Responses to Reviewer 2RUc (Part 1/3)**
>
> We sincerely thank the reviewer for the detailed comments. The suggestions of reviewer 2RUc make our work more perfect and help us a lot. Thank you!
>
> ------
>
> **W1.** We use an encoder with a BiMamba to process spatial structure information (i.e. protein-protein interaction network, subcellular location, and protein domains). The advantages of Mamba in processing spatial structure information are mainly reflected in its strong global modeling ability and linear complexity. Mamba, through its state-space model, can realize the perception and modeling of global information, so as to perform well in processing spatial structure information. Compared to the Transformer model, Mamba has lower computational complexity when dealing with long sequences. Mamba also performs well in processing multi-modal information. For example, in FusionMamba [1], the Mamba model is used in multi-modal image fusion tasks, which proves the effectiveness of Mamba in processing complex spatial structure information. Furthermore, in order to capture complex protein features more comprehensively in the model, we adopted a bidirectional scanning mechanism [2]. The BiMamba blocks we designed traverse features in both forward and reverse directions, enabling more accurate modeling of dependencies between proteins and between protein structures.
>
>  About BOW, this is a way of encoding protein structural features. We use BOW to encode protein subcellular localization and protein domain information, and the resulting features retain protein structural information. Just like when processing video signals, we need to convert video signals to digital signals through video encoding, the converted signal still has the main characteristics of the original video. Similarly, protein structure information is encoded so that the information is transformed into features that the model can learn, without affecting the main properties of the information.
>
>  In addition, **the BiMamba Block is not the core innovation of this work, even if replacing it with another block such as the Multihead Attention Block does not affect our main model structure.** We further demonstrated the effectiveness of BiMamba Block in the Protein Spatial Structure Information (PSSI) encoder by performing ablation experiments on key components of the encoder in **Appendix 6.2**. According to the table below, we can see that BiMamba Block is significantly more efficient in processing spatial structure information, **costing much less time (-5.122ms)** to predict a protein's function compared to Multihead Attention. We also find that the encoder using BiMamba Block as a component for modeling spatial structure features shows **significant advantages in MFO and CCO**. **Considering both inference efficiency and performance**, we decided to use an encoder with BiMamba Block as the component for spatial structure features modeling.
>
>
> | **Method**                       | $F_{max}$  |            |            | **m-AUPR** |            |            | **Cost Time (ms)** |
> | :------------------------------- | :--------: | :--------: | :--------: | :--------: | :--------: | :--------: | :----------------: |
> |                                  |  **BPO**   |  **MFO**   |  **CCO**   |  **BPO**   |  **MFO**   |  **CCO**   |                    |
> | PSSI Encoder+BiMamba  Block      |   0\.370   | **0\.240** | **0\.419** |   0\.244   | **0\.156** | **0\.371** |     **0\.539**     |
> | PSSI Encoder+Multihead Attention | **0\.430** |   0\.223   |   0\.350   | **0\.291** |   0\.154   |   0\.313   |       5\.661       |
>
>
>
> [1] Xie, Xinyu, et al. "Fusionmamba: Dynamic feature enhancement for multimodal image fusion with mamba", *IEEE Conference on Computer Vision and Pattern Recognition*, 2024.
>
> [2]Zhu, Lianghui, et al. "Vision mamba: Efficient visual representation learning with bidirectional state space model", arXiv preprint arXiv:2401.09417, 2024.
>
> ------

---

> ### Author Response · Authors · 2024-11-29
> **The Second Round of Responses to Reviewer 2RUc (Part 2/3)**
>
> **W2.** Thank you for your valuable reply. According to the reviewer's suggestion, we added more multimodal methods as comparison methods in **Table 1** of the revised paper. For example, the **Graph2GO** [3] method uses the same multimodal features as in this work (i.e. protein-protein interaction network, subcellular location, and protein domains). The experimental results show that our proposed BDGO model achieves superior performance, proving that multi-modal features play an important role in protein function prediction tasks. Furthermore, our BDGO model is superior to other multimodal methods, which proves the effectiveness of BDGO using bidirectional interaction and dynamic selection mechanisms.
>
>  The experimental results are shown as follows:
>
>
> | **Method**   | $F_{max}$  |            |            | **m-AUPR** |            |            |
> | :----------- | :--------: | :--------: | :--------: | :--------: | :--------: | :--------: |
> |              |  **BPO**   |  **MFO**   |  **CCO**   |  **BPO**   |  **MFO**   |  **CCO**   |
> | Graph2GO [3] |   0\.335   |   0\.196   |   0\.298   |   0\.237   |   0\.103   |   0\.215   |
> | BDGO (Ours)  | **0\.440** | **0\.282** | **0\.421** | **0\.332** | **0\.172** | **0\.392** |
>
>
>
>
> [3] Kunjie Fan, et al. “Graph2go: a multi-modal attributed network embedding method for inferring protein functions”, *GigaScience*, 9(8):giaa081, 2020.
>
> ------
>
> **Q2 -> 1).** Thank you for allowing us to clarify this issue. According to **rows 6 and 7 of Table 2 in the Ablation Studies Section**, we can see that, in our method, the performance of the sub-model using only sequence features is inferior to that of the sub-model using only spatial structure features. The experimental results are as follows. Here, w/o SP-F refers to removing spatial structure features from the input, and w/o SE-F indicates removing sequence features.
>
> | **Method** | $F_{max}$  |            |            | **m-AUPR** |            |            | **M-AUPR** |            |            |   **F1**   |            |            |  **ACC**   |            |            |
> | :--------- | :--------: | :--------: | :--------: | :--------: | :--------: | :--------: | :--------: | :--------: | :--------: | :--------: | :--------: | :--------: | :--------: | :--------: | :--------: |
> |            |  **BPO**   |  **MFO**   |  **CCO**   |  **BPO**   |  **MFO**   |  **CCO**   |  **BPO**   |  **MFO**   |  **CCO**   |  **BPO**   |  **MFO**   |  **CCO**   |  **BPO**   |  **MFO**   |  **CCO**   |
> | w/o SP-F   |   0\.216   |   0\.184   |   0\.265   |   0\.106   |   0\.102   |   0\.171   |   0\.104   | **0\.101** |   0\.112   |   0\.172   |   0\.174   |   0\.230   |   0\.152   |   0\.087   |   0\.156   |
> | w/o SE-F   | **0\.249** | **0\.272** | **0\.357** | **0\.118** | **0\.154** | **0\.212** | **0\.116** |   0\.082   | **0\.180** | **0\.181** | **0\.257** | **0\.307** | **0\.173** | **0\.128** | **0\.205** |
>
>
>  In order to make the comparison experiment more comprehensive, we have added the protein function prediction method (i.e., PredGO [4]) using the protein language model (PLM) as the comparison method in **Appendix 6.6**. According to the experimental results, it is evident that the BDGO model presented in this work outperforms other PLM-based methods. The experimental results are as follows:
>
> | **Method**  | $F_{max}$  |            |            | **m-AUPR** |            |            |
> | :---------- | :--------: | :--------: | :--------: | :--------: | :--------: | :--------: |
> |             |  **BPO**   |  **MFO**   |  **CCO**   |  **BPO**   |  **MFO**   |  **CCO**   |
> | PredGO [4]  |   0\.108   | **0\.455** |   0\.252   |   0\.058   | **0\.254** |   0\.183   |
> | BDGO (Ours) | **0\.440** |   0\.282   | **0\.421** | **0\.332** |   0\.172   | **0\.392** |
>
>
>
> [4] Zheng, Rongtao, Zhijian Huang, and Lei Deng. "Large-scale predicting protein functions through heterogeneous feature fusion", *Briefings in Bioinformatics*, 24(4): bbad243, 2023.
>
> ------

---

### Official Review · Reviewer_eqL1 · 2024-11-01

**Soundness:** 2
**Presentation:** 2
**Contribution:** 2
**Rating:** 3
**Confidence:** 3

**Summary:**

This paper presents BDGO, a novel multimodal approach for protein function prediction. The model includes a Bidirectional Interaction Module and a Dynamic Selection Module. The model incorporates the protein sequence and protein-protein interaction information. The proposed model is tested using the Gene Ontology prediction task.

**Strengths:**

1. The two modules (BInM and DSM) are proposed for cross-modal learning and multi-label selection.
2. The model Integrates multiple protein features (PPI network, subcellular location, protein domains and protein sequences)

**Weaknesses:**

1. The motivation for using PPI data is not clear. Many works used the protein structure, which may contain more information about protein function. Comparisons should be made with methods using structural information, such as DeepFRI and SaProt.
2. The testing task is limited. It may be helpful to see if the model can make reliable predictions on unannotated proteins.
3. The sequence encoder ProtT5 is already a pre-trained model. What is the reason for pre-training it again?
4. The effect of per-training is not clear.
5. There seem to be some hyperparameters in the model that are unclear.
6. In line 288, how are the groups of features generated?
7. In line 195, what is the "given surface"?

**Questions:**

See weakness

---

> ### Author Response · Authors · 2024-11-24
> **Response to Reviewer eqL1 (Part 1/2)**
>
> Thank you very much for your review. We respond to your concerns as follows. The modified parts in the revised paper are highlighted in blue.
>
> ------
>
> **W1/Q1.** We attach great importance to the comments of reviewers, reintroducing structure-based protein function prediction methods in Section **Introduction**, such as DeepFRI and SaProt. At the same time, the important contribution of DeepFRI and other structure-based methods in this field is discussed. In addition, we believe that PPI data can provide information about protein interactions, and there are many papers on protein function prediction based on PPI networks, such as GeneMANIA[1], Mashup [2], DeepNF[3], NetQuilt[4], DeepGO[5] and etc. These methods do not compare their results against structure-based approaches, as they focus solely on PPI data. Similarly, our method also incorporates PPI data and adheres to the same principle by not comparing against structure-based methods[6].
>
> ------
>
> **W2/Q2.** Thanks to the reviewer's comments, we are also curious about whether the model can make reliable predictions on unverified proteins. So we downloaded 272 unverified human protein data from the uniprot database. Among them, we removed proteins without PPI data. Finally, 136 proteins of BPO, MFO and CCO were obtained. The results of the test using our model are shown below. The experiment is supplemented in **Appendix Table 8 of Section 6.4**. At this time, this model has the highest results only in accuracy. This is because most of the 136 protein labels are not in our set 45 (BPO), 38 (MFO) and 35 (CCO), so they are mostly labeled 0. The results can be seen as follows:
>
> | **Method**         | **$F_{max}$** |         |         | **m-AUPR** |         |         | **M-AUPR** |         |         | **F1**  |         |         |  **ACC**   |            |            |
> | :----------------- | :-----------: | :-----: | :-----: | :--------: | :-----: | :-----: | :--------: | :-----: | :-----: | :-----: | :-----: | :-----: | :--------: | :--------: | :--------: |
> |                    |    **BPO**    | **MFO** | **CCO** |  **BPO**   | **MFO** | **CCO** |  **BPO**   | **MFO** | **CCO** | **BPO** | **MFO** | **CCO** |  **BPO**   |  **MFO**   |  **CCO**   |
> | BDGO               |    0\.440     | 0\.282  | 0\.421  |   0\.332   | 0\.172  | 0\.392  |   0\.171   | 0\.088  | 0\.257  | 0\.264  | 0\.263  | 0\.337  |   0\.331   |   0\.103   |   0\.210   |
> | BDGO (unannotated) |    0\.143     | 0\.025  | 0\.232  |   0\.015   | 0\.010  | 0\.101  |   0\.084   | 0\.041  | 0\.164  | 0\.024  | 0\.022  | 0\.115  | **0\.596** | **0\.559** | **0\.449** |
>
> ------
>
> **W3/Q3.** Thank you for giving us the opportunity to clarify the setting of ProtT5. We did not pre-train ProtT5. In our model, the ProtT5 model is frozen as part of our sequence encoder, and we pre-train the sequence encoder.
>
> ------

---

> > ### Author Response · Authors · 2024-11-24
> > **Response to Reviewer eqL1 (Part 2/2)**
> >
> > **W4/Q4.** Thanks for your question. As for pre-training reasons, to inject multimodal information, an encoder-decoder pre-training model is used, which learns the protein hidden embedding vector by reconstructing the original source features. In order to prove the effectiveness of pre-training, we added the pre-training ablation experiment in **Table 9 of Appendix Section 6.5**. The experimental results are as follows:
> >
> > | **Method**          | **$F_{max}$** |            |            | **m-AUPR** |            |            | **M-AUPR** |            |            |   **F1**   |            |            |  **ACC**   |            |            |
> > | :------------------ | :-----------: | :--------: | :--------: | :--------: | :--------: | :--------: | :--------: | :--------: | :--------: | :--------: | :--------: | :--------: | :--------: | :--------: | :--------: |
> > |                     |    **BPO**    |  **MFO**   |  **CCO**   |  **BPO**   |  **MFO**   |  **CCO**   |  **BPO**   |  **MFO**   |  **CCO**   |  **BPO**   |  **MFO**   |  **CCO**   |  **BPO**   |  **MFO**   |  **CCO**   |
> > | BDGO                |  **0\.440**   | **0\.282** | **0\.421** | **0\.332** | **0\.172** | **0\.392** | **0\.171** |   0\.088   | **0\.257** | **0\.264** | **0\.263** | **0\.337** |   0\.331   | **0\.103** |   0\.210   |
> > | BDGO $w/o$ pretrain |    0\.389     |   0\.176   |   0\.386   |   0\.195   |   0\.071   |   0\.225   |   0\.135   | **0\.105** |   0\.168   |   0\.248   |   0\.153   |   0\.297   | **0\.335** |   0\.144   | **0\.269** |
> >
> >
> > The experimental results show that the pre-training stage plays a key role in improving the model performance.
> >
> > ------
> >
> > **W5/Q5.** Thanks to the reviewers' suggestions, according to the author's suggestion, we have added the discussion of parameter setting in Implementation Details of **Section 3.1**. During fine-tuning, different hyperparameters are used for each aspect, with learning rates set to 3.6e-4, 8.6e-2, and 1.4e-4 for BPO, MFO, and CCO, respectively. The same pre-trained model serves as the feature extractor, and AdamW is used as the optimizer across all aspects. For BInM module, the cross-attention mechanism is configured with 8 heads, while other settings follow the default parameters in torch.nn. In DSM module, the temperature parameter $\tau$ is set to 1.
> >
> > ------
> >
> > **W6/Q6.** In our design, the features from the two branches (multimodal shared learning branch and multimodal interactive learning branch) are concatenated to form the input for the DSM. Each channel of concatenated features represents a feature group, so we have 6 feature groups. Since we have six groups of features, there are six experts. However, it is worth noting that DSM only choose one expert as the optimal feature classifier. A more detailed explanation can be found in **Section 2.2.2** **DYNAMIC SELECTION MODULE (DSM)**.
> >
> > ------
> >
> > **W7/Q7.** We thank the reviewer for pointing this out. We have changed the statement to “The spatial decoder rebuilds the given protein spatial information based on the hidden representations output by the encoder.” in **Section 2.1.1 SPATIAL ENCODER-DECODER**.
> >
> >
> >
> > [1] Mostafavi, Sara, et al. "GeneMANIA: a real-time multiple association network integration algorithm for predicting gene function." *Genome biology* 9 (2008): 1-15.
> >
> > [2] Cho, Hyunghoon, et al. "Compact integration of multi-network topology for functional analysis of genes." *Cell systems* 3.6 (2016): 540-548.
> >
> > [3] Gligorijević, Vladimir, et al. "deepNF: deep network fusion for protein function prediction." *Bioinformatics* 34.22 (2018): 3873-3881.
> >
> > [4] Barot, Meet, et al. "NetQuilt: deep multispecies network-based protein function prediction using homology-informed network similarity." *Bioinformatics* 37.16 (2021): 2414-2422.
> >
> > [5] Kulmanov, Maxat, et al. "DeepGO: predicting protein functions from sequence and interactions using a deep ontology-aware classifier." *Bioinformatics* 34.4 (2018): 660-668.
> >
> > [6] Lin, Baohui, et al. "A comprehensive review and comparison of existing computational methods for protein function prediction." *Briefings in Bioinformatics* 25.4 (2024): bbae289.

---

> > > ### Comment · Reviewer_eqL1 · 2024-11-26
> > >
> > > Thanks for the response. But I still have some concerns.
> > >
> > > **W1**: "These methods do not compare their results against structure-based approaches, as they focus solely on PPI data." but the focus of this submission is not on PPI data, but multi-modal protein data. And it is not a valid reason for not comparing against structure-based method. The methods you mentioned (GeneMANIA[1], Mashup [2], DeepNF[3], NetQuilt[4], DeepGO[5]) are not very recent and now the structure data is easy to acquire due to the Alphafold Database or other protein folding models. For multi-modal protein model, why PPI data over structure data? I am not very convinced of the reason for using PPI data.
> > >
> > > "At the same time, the important contribution of DeepFRI and other structure-based methods in this field is discussed." I did not find any content about DeepFRI.
> > >
> > > **W3**: Since you already use ProtT5, what is the purpose of pretraining the sequence encoder? The ProtT5 as the sequence encoder should be good enough. It seems that this is just for the whole framework to have a sequence pretraining component.

---

> > > > ### Author Response · Authors · 2024-11-29
> > > > **Responses to Reviewer eqL1**
> > > >
> > > > Thanks to the reviewer for the responses, please find our responses to your concerns below.
> > > >
> > > > **W1.** Thanks for the reminder, we have made modifications according to the reviewer's suggestions. In the **Section INTRODUCTION** of the revised paper (rows 54–58), we discussed structure-based protein function prediction methods and emphasized their significance. At the same time, we included discussions for DeepFRI and SaProt (rows 64–66 and rows 75–77).
> > > >
> > > > We agree with the reviewers’ opinion that the structures predicted by AlphaFold can aid protein function prediction. AlphaFold's structure prediction performance improves with updates, which benefits protein function prediction. In designing this experiment, we considered that AlphaFold-predicted structures might be affected by factors such as the cellular environment, which could lead to computational errors [1]. Therefore, in this work, we used experimentally determined PPI and other multimodal information, which prior research has shown to make significant contributions to protein function prediction. To further explore the performance differences between structure-based methods and BDGO (ours), we added a comparative experiment between BDGO and two structure-based methods (DeepFRI [2] and PredGO [3]) in **Appendix Section 6.6**, with the specific results shown below.
> > > >
> > > > | **Method**  | Focus                 | **$F_{max}$** |            |            | **m-AUPR** |            |            |
> > > > | :---------- | :-------------------- | :--------------------: | :--------: | :--------: | :--------: | :--------: | :--------: |
> > > > |             |                       | **BPO**                | **MFO**    | **CCO**    | **BPO**    | **MFO**    | **CCO**    |
> > > > | DeepFRI [2]     | Structure based       | 0\.362                 | **0\.461** | 0\.385     | 0\.308     | **0\.382** | 0\.360     |
> > > > | PredGO [3]     | Structure + PLM based | 0\.108                 | 0\.455     | 0\.252     | 0\.058     | 0\.254     | 0\.183     |
> > > > | BDGO (Ours) | Multi-modal based     | **0\.440**             | 0\.282     | **0\.421** | **0\.332** | 0\.172     | **0\.392** |
> > > >
> > > >
> > > > The experimental results show that BDGO outperforms other methods in BPO and CCO. We observe that BDGO and structure-based methods each have advantages in BPO, MFO, and CCO, and we infer that the multimodal and structural information used in this study may complement each other well.
> > > >
> > > >
> > > > References:
> > > >
> > > > [1] Pak, Marina A., et al. "Using AlphaFold to predict the impact of single mutations on protein stability and function.", Plos one 18(3): e0282689, 2023.
> > > >
> > > > [2] Vladimir Gligorijevi´c, et al. Structure-based protein function prediction using graph convolutional networks. Nature Communications, 12(1):3168, 2021.
> > > >
> > > > [3] Zheng, Rongtao, Zhijian Huang, and Lei Deng. "Large-scale predicting protein functions through heterogeneous feature fusion." Briefings in Bioinformatics 24(4): bbad243, 2023.
> > > >
> > > > ------
> > > >
> > > > **W3.** Thanks for your valuable feedback. Because ProtT5 is pre-trained on a broad dataset spanning multiple species, its performance on a specific task may not always meet expectations. Since our focus is on predicting human protein functions, we designed a Protein Sequence information (PSI) encoder to better fit our task. To achieve this, we froze the parameters of ProtT5 and connected it to our encoder for further pretaining.
> > > >
> > > > To further explore whether the PSI encoder is necessary, we added an additional experiment. In this experiment, we removed the PSI encoder and used the embeddings generated by ProtT5 as our sequence features. The experimental results show that, after removing the sequence encoder, the overall performance of our model decreased. **This further demonstrates that the PSI encoder we designed can make the model better adapted to our task.**
> > > >
> > > > | Method               | $F_{max}$ |           |       | m-AUPR |       |       | M-AUPR |       |       | F1    |       |       | acc   |       |       |
> > > > | -------------------- | ----- | ----- | ----- | ------ | ----- | ----- | ------ | ----- | ----- | ----- | ----- | ----- | ----- | ----- | ----- |
> > > > |                      | **BPO** | **MFO** | **CCO** | **BPO**   | **MFO**   | **CCO**   | **BPO**   | **MFO**   | **CCO**   | **BPO**   | **MFO**   | **CCO**   | **BPO**   | **MFO**   | **CCO**   |
> > > > | BDGO (Ours)           | **0.440** | **0.282** | **0.421** | **0.332** | **0.172** | **0.392** | **0.171** | 0.088 | **0.257** | **0.264** | **0.263** | **0.337** | **0.331** | 0.103 | **0.210** |
> > > > | BDGO w/o PSI encoder | 0.346 | 0.172 | 0.367 | 0.250  | 0.082 | 0.252 | 0.157  | **0.098** | 0.198 | 0.234 | 0.156 | 0.292 | 0.247 | **0.113** | 0.210 |

---

### Official Review · Reviewer_JcGD · 2024-11-01

**Soundness:** 3
**Presentation:** 3
**Contribution:** 4
**Rating:** 8
**Confidence:** 4

**Summary:**

The paper "Improving Multimodal Protein Function Prediction Using Bidirectional Interaction and Dynamic Selection Mechanisms" introduces the BDGO model, aimed at enhancing multimodal protein function prediction through a combination of spatial and sequence features. The authors propose two primary modules: the Bidirectional Interaction Module (BInM) for interactive learning between multimodal features, and the Dynamic Selection Module (DSM) for optimizing hierarchical multi-label classification. Comprehensive experiments on human datasets reveal that BDGO outperforms current multimodal-based methods, such as CFAGO, regarding F-max and m-AUPR. The paper also includes t-SNE visualization and Davies-Bouldin score analyses to validate the effectiveness of the features extracted by BDGO.

**Strengths:**

- Originality: The BDGO model is innovative in its approach to bidirectional interaction. It addresses challenges in multimodal protein function prediction by integrating spatial and sequence features through interactive and dynamic learning.
- Quality: The experiments are thorough, using various metrics (e.g., F-max, m-AUPR) and comparisons against state-of-the-art methods. The inclusion of ablation studies adds depth to the validation of BDGO’s components.
- Clarity: The paper clearly defines each module's role and the experimental setup. Figure 1 effectively illustrates the BDGO architecture, aiding comprehension of the methodology.
- Significance: BDGO's improvements in protein function prediction highlight its potential as a valuable tool in computational biology, particularly in handling complex, multimodal data.

**Weaknesses:**

- Methodological Details: The rationale behind specific design choices, such as the number of heads in cross-attention or layer normalization parameters, could be elaborated upon to aid reproducibility.
- Generalizability: The paper primarily focuses on human datasets. Extending the approach to datasets from other species could provide insight into BDGO’s generalizability.

**Questions:**

1. Could the authors clarify the parameter selection process for the BInM and DSM modules? Understanding this would improve insight into BDGO's adaptability.
2. Did the authors explore any augmentation techniques for protein features? Such techniques could potentially address overfitting in low-sample domains.
3. Would the BDGO model benefit from additional layers in the DSM, particularly when dealing with very large protein function datasets?

---

> ### Author Response · Authors · 2024-11-24
> **Response to Reviewer JcGD**
>
> We are greatly encouraged by your insightful comments, e.g. an innovative method, valuable tool, and extensive experiment. In response to your concerns, we offer the following explanation. The modified parts in the revised paper are highlighted in blue.
>
> ------
>
> **W1.** Thanks for the reminder of the reviewer, we have added the specific settings of parameters in Implementation Details in **Section 3.1** of the revised paper.
>
> ------
>
> **W2.** Thanks to the reviewers' suggestions, we will continue to explore protein function prediction methods in other species, such as mice, in future studies.
>
> ------
>
> **Q1.** According to the author's suggestion, we have added the discussion of parameter setting in Implementation Details of **Section 3.1**. During fine-tuning, different hyperparameters are used for each aspect, with learning rates set to 3.6e-4, 8.6e-2, and 1.4e-4 for BPO, MFO, and CCO, respectively. The same pre-trained model serves as the feature extractor, and AdamW is used as the optimizer across all aspects. For BInM module, the cross-attention mechanism is configured with 8 heads, while other settings follow the default parameters in torch.nn. In DSM module, the temperature parameter $\tau$ is set to 1.
>
> ------
>
> **Q2.** Thanks for the comments. The pre-training method can effectively enhance the expression of protein features, which is more prominent when combined with multi-modal information. Our study uses an encoder-decoder architecture that combines PPI, subcellular localization, protein domain, and sequence information to improve feature representation through the injection of multimodal knowledge, thereby alleviating overfitting problems in low-sample domains. In addition, large protein models (such as ESM, ProtTrans, etc.), *which are also utilized in our study,* extract deep features from massive sequences through self-supervised learning. These models perform better in data scarcity scenarios than traditional feature designs.
>
> ------
>
> **Q3.** Thanks to the reviewer for inspiring us. When working with very large protein function data sets, there may be potential benefits to adding some extra layers to the DSM. As far as we know, the directions mentioned by the reviewers are highly relevant to some research work in recent years in multiple expert fields [1, 2, 3]. Therefore, dynamic allocation of expert modules is an effective strategy worth exploring. We believe that further improving the expert module on the basis of the existing DSM model may be a research direction worth exploring. In the future, we will actively attempt to implement these strategies on larger data sets and promptly begin related research to further validate their effects. Once again, we sincerely thank the reviewer for their innovative and valuable suggestions.
>
>
>
> [1] Xue, Zihui, et al. "Dynamic multimodal fusion." *Proceedings of the IEEE/CVF Conference on Computer Vision and Pattern Recognition*, 2023.
>
> [2] Zhang, Qingyang, et al. "Provable dynamic fusion for low-quality multimodal data." *International conference on machine learning*. PMLR, 2023.
>
> [3] Han, Xing, et al. "Fusemoe: Mixture-of-experts transformers for fleximodal fusion." *arXiv preprint arXiv:2402.03226* (2024).

---

### Official Review · Reviewer_SK3h · 2024-11-03

**Soundness:** 1
**Presentation:** 1
**Contribution:** 2
**Rating:** 5
**Confidence:** 4

**Summary:**

The authors propose a connectionist approach to jointly exploit different sources of information (they term this multimodal) on proteins ranging from sequence to pairwise interactions from cellular localisation to domain composition, to tackle the problem of protein function prediction.

The paper's contributions are: 1) differently from current multimodal approaches that primarily rely on information fusion mechanisms they consider the potential 'complementarity' between different modalities and 2) they develop an approach so that each modality not only influences the processing of other modalities but also 'obtains information' from them, thereby enhancing the overall understanding capability and 3) since protein function prediction is essentially a complex hierarchical multi-label classification problems they propose an approach to dynamically select the optimal feature combination for 'fitting more diverse' protein functions.

**Strengths:**

The idea of effectively iterating multiple and diverse sources of information is of interest.

**Weaknesses:**

1. The manuscript would benefit from enhanced clarity: the approach is made up of a large number of components, requiring careful attention to clearly explain the purpose of each part and how they relate to one another in an easy-to-follow manner.
2. each stated contribution is 1) not formally defined; 2) a formal way to measure its efficacy is not presented and 3) clear empirical experiments to show that the contribution is effective are not offered.
3. The results are presented without any measure of dispersion (e.g., standard deviation), making it difficult to determine whether the comparisons are significative.

**Questions:**

1. which is the authors contribution in the design of the individual modules is not clear: a) please provide citations for BiMamba or claim to be its authors, b) is Bidirectional Interaction Module (BINM) originally proposed for the first time in the current work? please explicitly state it if this is the case. c) is Dynamic Selection Module (DSM) the Mixture-of-Experts (MoE) from Masoudnia & Ebrahimpour, 2014, or does it introduces novelties (since <<Different from the traditional MoE system that weighted fuses all branches, our hard gating network selects one of the branches for calculation, which makes the model adapt to the prediction of the large-number and complex protein functions.>>)
2. the author offer 3 contributions:  1) not only information fusion but also a way to exploit potential 'complementarity' between different modalities: how do you show that is the complementarity that is being captured? can you design an artificial case where one can influence the level of complementarity and show that this approach can exploit it better than methods that only rely on information fusion? 2) the BINM should allow to 'obtain information' from each modality, thereby enhancing the overall understanding capability: can an artificial case be designed so that a notion of cross-talk between modalities can be manipulated and can we show that BINM can exploit that? 3) DSM should allow to dynamically select the optimal feature combination for 'fitting more diverse' protein functions: can we design an experiment to show how only some expert are used and in which contexts are they selected?
3. when comparing approaches please consider using critical difference diagrams (https://scikit-posthocs.readthedocs.io/en/latest/generated/scikit_posthocs.critical_difference_diagram.html) to report if the results are significative: a) one diagram could complement table 1 considering repeated experimental results for all performance measures and all tasks (please offer the acronym explanation for BPO MFO CCO in the text) at the same time, and could be used to answer the question: does BDGO significantly outperforms other approaches (on all task and on all measures)? b) another diagram could complement Fig. 6 and could be used to answer the question: is any ablated element contributing significantly to the performance improvement?

---

> ### Author Response · Authors · 2024-11-24
> **Response to Reviewer SK3h (Part 1/2)**
>
> We sincerely thank the reviewer for the comments and suggestions to our manuscript. We have carefully studied these comments and revised our manuscript based on them. Please find our responses to the comments below. The modified parts in the revised paper are highlighted in blue.
>
> ------
>
> **W1.** Thank you for giving us the opportunity to clarify the purpose and relationship of each component.
>
> Our proposed method efficiently captures multimodal information of proteins through a two-step training strategy . In the pre-training stage, we use the encoder-decoder model to learn and inject multimodal knowledge. For spatial features including PPI, subcellular location, and protein domains, a spatial encoder-decoder model using the BiMamba blocks is introduced in this stage. To mine sequence features including protein sequences, we design a sequence encoder-decoder model based on the Transformer blocks for pre-training. Then, during our BDGO model training phase, we integrate and learn features from multimodal information. The proposed model is primarily divided into two major branches: one is the multimodal shared learning branch, and the other is the multimodal interactive learning branch. Protein data are processed through these multiple branches to generate several sets of features, which serve as inputs for our well-designed hard gating network. Finally, the model dynamically selects the optimal features for each protein to enhance performance in protein function prediction.
>
> Following your suggestions, we have added detailed explanations in Section **METHODOLOGY** and revised the descriptions in **Section 2.2** to enhance clarity and coherence.
>
> ------
>
> **W2.** Thank you for your comments. Regarding formal definitions, we have thoroughly explained and defined each component in detail in the **METHODOLOGY** section. As for efficacy measurement, we have analyzed the effect of each component in the **Ablation Studies** section. The corresponding experimental results are presented in **Table 2**.
>
> ------
>
> **W3.** Thanks for the suggestions. In order to increase the statistical significance of the experimental results, we conducted five tests for each model and calculated the variation range of the test results. The supplementary experimental results of BDGO have been added to **Table 1**, which can be seen as follows:
>
> | **Aspect** | **$F_{max}$** | **m-AUPR**  | **M-AUPR**  |   **F1**    |   **ACC**   |
> | :--------- | :-----------: | :---------: | :---------: | :---------: | :---------: |
> | BPO        |  0.440±0.013  | 0.332±0.007 | 0.171±0.004 | 0.264±0.007 | 0.331±0.012 |
>
>
> | **Aspect** | **$F_{max}$** | **m-AUPR**  | **M-AUPR**  |   **F1**    |   **ACC**   |
> | :--------- | :-----------: | :---------: | :---------: | :---------: | :---------: |
> | MFO        |  0.282±0.038  | 0.172±0.014 | 0.088±0.012 | 0.263±0.036 | 0.103±0.040 |
>
>
> | **Aspect** | **$F_{max}$** | **m-AUPR**  | **M-AUPR**  |   **F1**    |   **ACC**   |
> | :--------- | :-----------: | :---------: | :---------: | :---------: | :---------: |
> | CCO        |  0.421±0.013  | 0.392±0.012 | 0.257±0.011 | 0.337±0.018 | 0.210±0.041 |
>
> ------

---

> ### Author Response · Authors · 2024-11-24
> **Response to Reviewer SK3h (Part 2/2)**
>
> **Q1.** Thanks for the comments. In this work, we designed several modules for multimodal feature interaction and feature selection to improve the protein function prediction ability of the model.
>
> **As for BiMamba Block:** Inspired by the bidirectional scanning mechanism proposed in the Vision Mamba [1] method from the field of computer vision, our proposed BiMamba Block innovatively applies the bidirectional scanning mechanism to the learning of spatial features of proteins. The modified parts in BiMamba Block of **Section 2.1.1** are highlighted in blue.
>
> **As for Bidirectional Interaction Module (BInM):** We are the first to propose a BInM for multimodal protein feature interactions. Multimodal BInM enhances the understanding and learning ability of complex patterns by combining spatial information and sequence information of different modes. BInM consists of two-branch cross-attention blocks, and mining the interrelation of multimodal data through two-way information interaction. The modifications made to BInM in **Section 2.2.1** are indicated in blue.
>
> **As for Dynamic Selection Module (DSM):** Where our DSM differs from traditional MoE is that our experts only need to acquire one feature, whereas traditional MoE uses a combination of features. This is because the input features of our DSM are multi-channel features that have been interactively integrated. In order to ensure the best interaction effect, we only learn one channel feature for each expert, and select the best expert through DSM. According to experimental results, we found that the effect of experts who selected only one feature was better than that of experts who selected multiple features. The experimental results are supplemented in Appendix **Table 5 in Section 6.1.2**. Additionally, we have made modifications to the description of DSM in **Section 2.2.2** to provide further clarity on its design and functionality.
>
> ------
>
> **Q2.**
>
> **1\)** To demonstrate that we capture the complementarity between sequence and spatial features, we supplement a feature ablation experiment in which we retain only spatial or sequence features. It can be found that features cannot interact when only spatial or sequential features are used, and the model performs poorly. The complementarity of the characteristics of various modes is proved. The experimental results are shown in **row 6 and 7 of Table 2**.
>
> As for the artificial case mentioned by reviewer, we have compared our method (BDGO) with one that relies solely on information fusion (MSL-Branch) in ablation experiments. With the addition of MIL-Branch for feature information interaction, BDGO gets an overall performance boost. The experimental results are shown in **row 1, 2, and 3 of Table 2**. In addition, we have restated this comparison in **ABLATION STUDIES**  for further clarification.
>
>  **2\)** Thank the reviewers for their comments. We designed an ablation experiment in which we changed the feature interactions in the BInM module. As shown in **Table 3 in Appendix Section 6.1.1**, BDGO-BInM-0 and BDGO-BInM-1 represent removing the bottom and top cross-attention module from BInM.
>  BDGO-BInM-0 is calculated as :
>  $F_c^{(1)} = softmax(\mathcal{Q}^{(1)}(F_b^{(1)}) \otimes \mathcal{K}^{(2)}(F_b^{(2)})^T) \otimes \mathcal{V}^{(2)}(F_b^{(2)})$, $F_c^{(2)} = F_b^{(2)}$.
>  BDGO-BInM-1 is calculated as :
>  $F_c^{(1)} = F_b^{(2)}$, $F_c^{(2)} = softmax(\mathcal{Q}^{(2)}(F_b^{(2)}) \otimes \mathcal{K}^{(1)}(F_b^{(1)})^T) \otimes \mathcal{V}^{(1)}(F_b^{(1)})$.
>  As can be seen in **Table 3**, the performance of BDGO decreased after changing the interaction mechanisms. The advantages of our BInM feature-interaction design are proved.
>
> **3\)** The experts in our DSM only need to acquire one feature, and instead adopt a combination of features. We supplement an experiment in the DSM in which experts select multiple features, in which we find that experts who select multiple features perform less well than experts who select only a single feature. The experimental results are shown in **Table 5 in Appendix Section 6.1.2**.
>
> ------
>
> **Q3.** Thanks to the reviewer's suggestions, we have made corresponding modifications in the article.
>
> **a)** In Section **INTRODUCTION**, we have added the abbreviations for "Gene Ontology" (GO), "Biological Process Ontology" (BPO), "Molecular Function Ontology" (MFO), and "Cellular Component Ontology" (CCO).
>
> **b)** In **Appendix 6.8** **CRITICAL DIFFERENCE DIAGRAMS FOR STATISTICAL COMPARISON**, we have added the critical difference diagrams for the comparison with other methods and the ablation experiments. Through critical difference diagrams, we can see that BDGO is significantly better than other methods, which proves the superiority of our method design.
>
>
>
> [1] Zhu, Lianghui, et al. "Vision mamba: Efficient visual representation learning with bidirectional state space model." *arXiv preprint arXiv:2401.09417* (2024).

---

> > ### Comment · Reviewer_SK3h · 2024-12-02
> >
> > If it is at all possible, it would be informative to report a single CDC by considering all reasons of variability jointly:
> > 1) simply out together the experimental results for all performance measures and all tasks at the same time to answer the question: does BDGO significantly outperforms other approaches (on all task and on all measures)?
> > 2) is each ablated element contributing significantly to the performance improvement on all tasks for all performance measures?

---

> > > ### Author Response · Authors · 2024-12-04
> > > **The Second Round of Responses to Reviewer SK3h**
> > >
> > > We sincerely thank the reviewer for the valuable feedback. We apologize for not fully understanding the abbreviation "CDC", but we infer that critical difference diagrams (CDD) can be used to address the two important suggestions. Although we have completed the requested visualization, unfortunately, the pdf upload time has passed. We will include these visualizations in the appendix for future versions. The related image can be viewed from the following link: https://anonymous.4open.science/r/Supplementary-data-34F8/CDD_all.png
> > >
> > >  According to the CDD results, our model BDGO ranks first and significantly outperforms other methods across all tasks and evaluation metrics. However, there is no statistically significant difference between BDGO and CFAGO. This may be due to their small score difference, which makes it hard to detect a significant statistical difference.
> > >
> > >  Additionally, the CDD results show that ablation methods such as w/o Gate, w/o BInM, MILB, MSLB, w/o SP-F, and w/o SE-F perform significantly worse than BDGO. This highlights the importance of each component in our model, as they all contribute significantly to performance across all tasks and metrics.
> > >
> > >  In response to the reviewers' earlier suggestions, we have added critical difference diagrams in **Appendix 6.8** to analyze BDGO's advantages and the contributions of its components. Overall, BDGO demonstrates superior performance across all tasks. Notably, in **CCO**, BDGO achieves significant improvements in both Fmax and M-AUPR compared to other methods. Similarly, for **MFO**, BDGO outperforms competing approaches significantly in Fmax. Specifically, based on the comparative results in **Table 1**, BDGO achieves at least a 0.046 and 0.055 improvement in Fmax for MFO and CCO, respectively.
> > >
> > >  For the ablation studies, our **critical difference diagrams** clearly show that removing individual components reduces BDGO's performance. From **Table 2**, we see that excluding the **BInM** module significantly lowers results for MFO and CCO, showing its key role in these tasks. Similarly, removing the **DSM** module decreases Fmax by 0.036 for **BPO**, 0.115 for **MFO**, and 0.048 for **CCO**, proving its importance across all three aspects. Additionally, neither the **MSLB** nor the **MILB** branch alone overmatches BDGO's overall performance, showing that combining multimodal features from multiple branches is essential for better predictions.
> > >
> > >  In summary, BDGO demonstrates significant advantages in **MFO** and **CCO** compared to other methods, with each of its components contributing distinct strengths across different tasks.

---

### Comment · Area_Chair_vWrZ · 2024-11-26
**appreciate if you could respond to the authors' rebuttal**

Dear reviewers, this paper has split reviews, and thus, we really need your help to engage. I would really appreciate it if you could share your responses to the author's rebuttal -- whether that solves your concern or not.

Thanks, AC vWrZ

---

### Note · Authors · 2025-01-17

I have read and agree with the venue's withdrawal policy on behalf of myself and my co-authors.